# Shared behavioural impairments in visual perception and place avoidance across different autism models are driven by periaqueductal grey hypoexcitability in Setd5 haploinsufficient mice

**Laura E. Burnett[1]¤, Peter Koppensteiner[1], Olga Symonova[1], Tomás Masson[1], Tomas Vega-Zuniga[1], Ximena Contreras[1], Thomas Rülicke[2], Ryuichi Shigemoto[1], Gaia Novarino[1], Maximilian Joesch[1]***

**1** Institute of Science and Technology Austria, Klosterneuburg, Austria, **2** Department of Biomedical Sciences and Ludwig Boltzmann Institute for Hematology and Oncology, University of Veterinary Medicine, Vienna, Austria

¤ Current address: Janelia Research Campus, Howard Hughes Medical Institute, Ashburn, Virginia, United States of America

* maxjosch@ist.ac.at

**Data Availability Statement:** The mass spectrometry proteomics data have been deposited

## Abstract

Despite the diverse genetic origins of autism spectrum disorders (ASDs), affected individuals share strikingly similar and correlated behavioural traits that include perceptual and sensory processing challenges. Notably, the severity of these sensory symptoms is often predictive of the expression of other autistic traits. However, the origin of these perceptual deficits remains largely elusive. Here, we show a recurrent impairment in visual threat perception that is similarly impaired in 3 independent mouse models of ASD with different molecular aetiologies. Interestingly, this deficit is associated with reduced avoidance of threatening environments—a nonperceptual trait. Focusing on a common cause of ASDs, the *Setd5* gene mutation, we define the molecular mechanism. We show that the perceptual impairment is caused by a potassium channel (Kv1)-mediated hypoexcitability in a subcortical node essential for the initiation of escape responses, the dorsal periaqueductal grey (dPAG). Targeted pharmacological Kv1 blockade rescued both perceptual and place avoidance deficits, causally linking seemingly unrelated trait deficits to the dPAG. Furthermore, we show that different molecular mechanisms converge on similar behavioural phenotypes by demonstrating that the autism models *Cul3* and *Ptchd1*, despite having similar behavioural phenotypes, differ in their functional and molecular alteration. Our findings reveal a link between rapid perception controlled by subcortical pathways and appropriate learned interactions with the environment and define a nondevelopmental source of such deficits in ASD.

at the ProteomeXchange Consortium via the PRIDE partner repository with the data set identifier PXD051022. The data used in the analysis can be found in the ISTA data repository DOI: 10.15479/AT:ISTA:1585. The code used to generate the results is available at: https://zenodo.org/records/1110588.

**Funding:** This work was supported by a European Research Council Starting Grant 756502 (MJ). The funders had no role in study design, data collection and analysis, decision to publish, or preparation of the manuscript.

**Competing interests:** The authors have declared that no competing interests exist.

**Abbreviations:** AAV, adeno-associated virus; ACSF, artificial cerebrospinal fluid; AIS, axon initial segment; ASD, autism spectrum disorder; BCA, bicinchoninic acid assay; BLA, basolateral amygdala; BSA, bovine serum albumin; ChR2, channelrhodopsin-2; dmSC, deep medial SC; dPAG, dorsal periaqueductal grey; DS, direction selective; DSI, direction selectivity index; EPSC, excitatory postsynaptic current; FDR, false discovery rate; ID, inflection depth; idSC, intermediate/deep superior colliculus; IP, intraperitoneal; LED, light-emitting diode; LER, looming escape response; LP, lateral posterior; PAG, periaqueductal grey; PFA, paraformaldehyde; PLR, pupillary light reflex; PMSF, phenylmethylsulfonyl fluoride; RMP, resting membrane potential; RT, room temperature; SC, superior colliculus; SD, standard deviation; sEPSC, spontaneous excitatory postsynaptic current; Setd5, SET-domain containing 5; SGS, stratum griseum superficiale; SNR, signal-to-noise ratio; SO, stratum opticum; sSC, superficial superior colliculus; WT, wild-type.

## Introduction

Autism spectrum disorders (ASDs) are conditions characterised by challenges with social interactions and repetitive behaviours, reflected in inadequate responses to others' mental states and emotions [1,2]. These alterations in social cognition co-occur with disordered sensory processing [3], a widespread yet often overlooked feature across ASD observed in every sensory modality [4]. In the visual domain, affected individuals frequently exhibit difficulties with visual attention and hyper- or hyposensitivity to visual stimuli [5], and mounting evidence suggests that the circuits involved in visual information processing are disrupted [6]. For example, atypical neuronal responses to faces and looming stimuli have been observed in individuals with ASD [7–9]. Such sensory deficits directly affect visually guided behaviours such as gaze control, which emerges at a few months of age [10] and forms a prominent diagnostic feature [11]. Changes in gaze dynamics to subliminal stimuli [12] indicate that neuronal circuits mediating subconscious visual responses are affected, pointing towards subcortical circuits, particularly the superior colliculus (SC) [6,13,14].

Deficits in sensation are seemingly separate phenomena from the more studied cognitive impairments in ASD, believed to arise from cortical malfunctions [15–17]. However, their close association indicates the possibility of a common underlying thread, particularly because their severity is associated with the strength and expression of other ASD traits [18–20]. Such evidence has recently implicated SC pathway impairments with atypical sensory and cognitive processing [6,14]. Given this evidence, we decided to directly explore the relationship between sensation and cognition in ASD within an innate, robust, and reproducible sensorimotor transformation known to be mediated by the SC—the looming escape response (LER) [21,22]. Using *Setd5* [23] as a case study, we show that the LER is impaired, not due to direct sensory or motor deficits. While all animals immediately detect the looming stimulus and can respond vigorously to a threat, they require longer to initiate the escape response than their wild-type siblings and do not form an appropriate aversion to the threat area. These 2 behavioural traits are highly correlated. The stronger the perceptual deficits, the weaker the avoidance. We further show that these behavioural correlations are also present in aetiologically distinct ASD mouse models (*Cul3* [24] and *Ptchd1* [25]), indicating that these various molecular dysfunctions converge on common behavioural deficits. In *Setd5*, these deficits emerge through changes in the intrinsic excitability due to an increased potassium channel conductance of neurons in the dorsal periaqueductal grey (dPAG), a structure known for commanding escape responses and receiving direct input from the SC [26,27]. Rescuing this hypoexcitability phenotype in the dPAG in adult mice recovers both the perceptual and place avoidance deficits, linking a sensorimotor disorder via a defined molecular mechanism to cognitive dysfunction. Our results show that dPAG dysfunctions can be instrumental in the emergence of symptoms associated with ASDs and that some of the dysfunctions are not developmental, opening a path for their targeted treatment.

## Results

### Delayed looming escape responses

To investigate whether subcortical visuomotor transformations could be affected by genetic mutations associated with ASD, we evaluated the behavioural responses to the innate LER paradigm. LER is largely independent of cortical input, requiring the SC and its downstream targets to initiate appropriate responses. For this purpose, we tested a genetic haploinsufficiency ASD mouse model affecting the *SET-domain containing 5 (Setd5)* [23], a histone-associated protein found in approximately 1% of patients with intellectual disability and ASD [23]. We

subjected mutants (i.e., *Setd5*$^{+/−}$) and their wild-type (WT) siblings to our LER paradigm, where mice would automatically trigger a sequence of 5 consecutive looms upon entering a threat zone (Fig 1A and 1B; see **Materials and methods**). For all experiments, we only used animals without any craniofacial abnormalities, tooth displacement, or eye abnormalities, which are described to sometimes occur in our *Setd5* mutant animals [23]. As previously reported, *Setd5*$^{+/−}$ animals were, on average, slightly smaller than their WT siblings [23]. Please note that other models for the same gene show more drastic abnormalities not observed in our hands [28]. During the acclimatisation period, all animals showed similar exploratory behaviour and dynamics (S1A–S1G Fig), and seemingly similar behavioural responses, escaping to the shelter upon stimulus presentation (S1 Video). However, behavioural differences between the genotypes became evident when observing their reaction times (Fig 1C). While WT

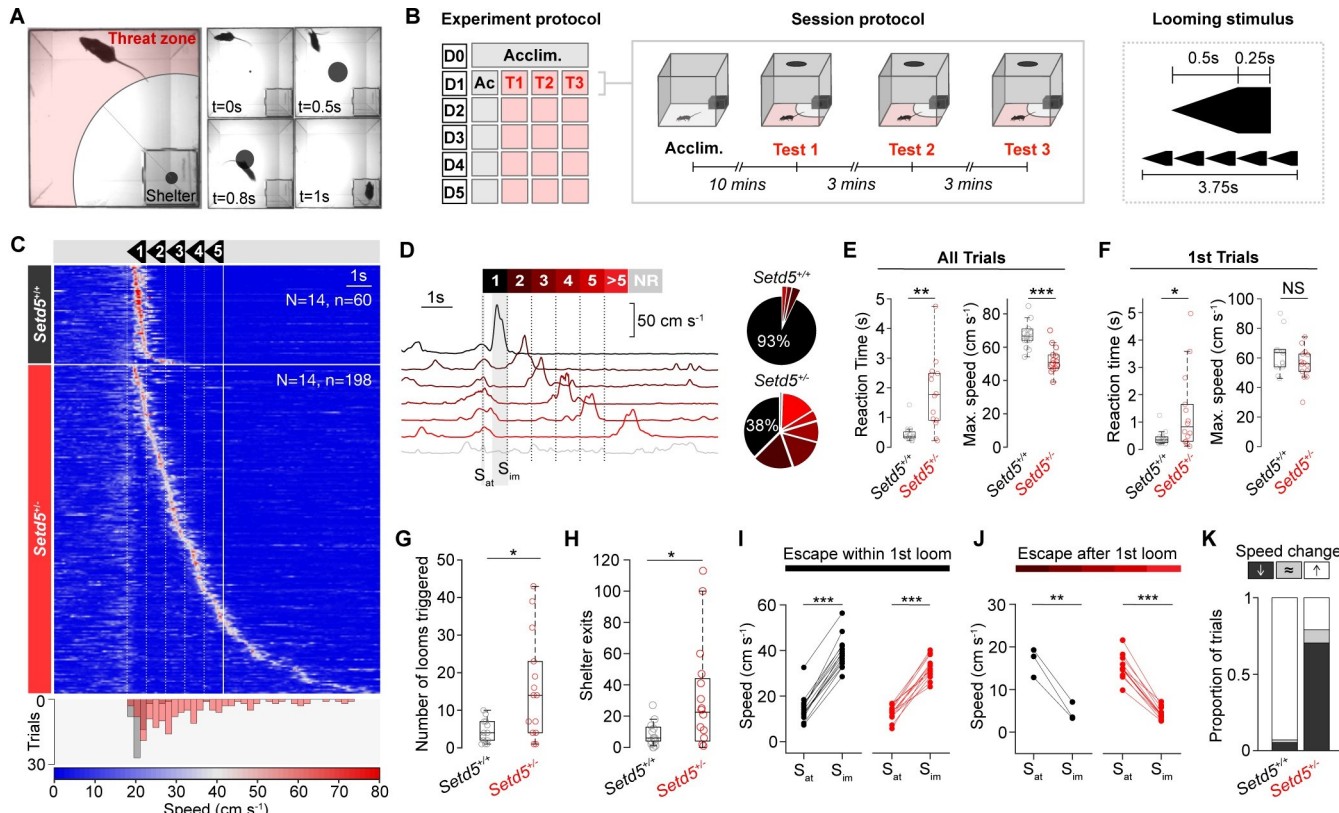

**Fig 1. ASD mice exhibit delayed and less vigorous looming escape responses.** (**A**) LER paradigm showing the shelter's location and the threat zone (left) and LER example (right). (**B**) Paradigm schematic. Day 0 (D0) was used for acclimatisation. D1-D5 consisted of an acclimatisation period (grey) followed by 3 LER tests (red). The looming stimulus consisted of 5 consecutive looms (right). (**C**) Raster plot of mouse speed during LER (white, dotted vertical lines denote the start of each loom; solid white line denotes the end of the stimulus) for *Setd5*$^{+/+}$ (upper, *n* = 14, 60 trials) and *Setd5*$^{+/−}$ (lower, *n* = 14, 198 trials), sorted by reaction time. Bottom, distribution of reaction times for all *Setd5*$^{+/+}$ (black) and *Setd5*$^{+/−}$ (red) trials (*p* < 0.001, two-sample Kolmogorov–Smirnov test). (**D**) Left, example trials based on whether the mouse responds within 1 of the 5-loom stimuli, after the fifth (>5), or not at all (NR, no response) from one *Setd5*$^{+/−}$ mouse. Grey shaded areas represent the frames used to calculate the speed at stimulus onset (S$_{at}$) and the immediate response speed (S$_{im}$). Right, proportion of escapes to loom presentations. (**E**) Total looms triggered across the 5 test days (*Setd5*$^{+/+}$, 4.36 looms; *Setd5*$^{+/−}$, 16 looms, *P* = 0.013). (**F**) Total shelter exits across the 5 test days (*Setd5*$^{+/+}$, 6.0; *Setd5*$^{+/−}$, 24.5, *P* = 0.019). (**G**) Average reaction time (left) and maximum escape speed (right) per animal (reaction time, *P* = 0.001; max. escape speed, *P* < 0.001). (**H**) As (**G**), but only for the very first loom presentation (reaction time, *P* = 0.046; max. escape speed, *P* = 0.166). (**I**) Average immediate speed change following the stimulus presentation for all trials where the mice escape within the first loom presentation (left, *Setd5*$^{+/+}$, *n* = 14, black, *p* < 0.001; right, *Setd5*$^{+/−}$, *n* = 14, red, *p* < 0.001). S$_{at}$ is the mean speed of the animal ±50 ms of stimulus onset, and S$_{im}$ is the mean speed of the animal 300–800 ms after stimulus onset. (**J**) As (**I**) but for trials where the mice escape during or after the second loom (left, *Setd5*$^{+/+}$, *n* = 3, black, *p* = 0.007; right, *Setd5*$^{+/−}$, *n* = 11, red, *p* < 0.001). (**K**) Proportion of response types per genotype (X$^2$ = 103.9, *p* < 0.001, X$^2$ test of independence). *P*-values: Wilcoxon's test, *p*-values: paired *t* test, unless specified. The data underlying this figure can be found in S1 Data.

siblings responded robustly to the first stimulus presentation, their heterozygous siblings required, on average, more repetitions to elicit an escape response (Fig 1D), triggered more looms (Fig 1E), made more shelter exits (Fig 1F), had longer reaction times, and escaped with less vigour, reaching lower maximum speeds [29] (Fig 1G). The differences across genotypes were independent of the location, speed of the animal at stimulus onset or heading angle (S1H–S1J Fig), or sex (Fig 1C–1J, females; and S1L–S1N Fig, males). The difference in response reaction time is an intrinsic and not an adaptive property, since it is present from the first loom presentation the animal encounters (Fig 1H and 1G, left). However, the escape vigour of the $Setd5^{+/-}$ animals during these first-ever encountered loom presentations was indistinguishable from their WT siblings (Fig 1H, right). This indicates that the escape delay differences are not due to intrinsic motor problems but emerge due to perceptual decision-making deficits—the process of extracting and integrating sensory information to initiate a subsequent action. These results highlight the LER as a robust behavioural assay to probe perceptual impairments and their underlying circuit dysfunctions in visuomotor processing in ASD.

To test whether the delay in initiating an escape to the loom was due to the mutant mice simply detecting the stimulus later than their WT siblings, we examined the immediate change in the speed of the animals from the time of the first loom, selecting only the trials where the mice performed an escape to the shelter at any point during the looming events. For trials where the animals generated an escape within the first loom presentation, both WT and mutant animals increased their speed significantly upon stimulus onset (Fig 1I). For trials where the mice did not generate an escape within the first loom, which was the majority of trials for mutant animals but only a small fraction of trials for WT animals (Fig 1D), the mice significantly reduced their speed in the time immediately following the stimulus onset (Fig 1J). This demonstrates that $Setd5^{+/-}$ animals detect and respond to the stimulus within a similar time frame as their WT siblings but preferentially perform a locomotor arrest instead of an escape response (see S1–S3 Videos). This arrest behaviour is more reminiscent of risk assessment behaviour as previously shown for loom stimuli of different contrast [29] rather than the defensive freezing response characterised by the sustained cessation of all movement, since the animals were still performing small movements of their head and upper body. Indicating that the animals are using this time to perform ongoing threat evaluation. This suggests that the delay in LER arises from difficulties in either evaluating the threat level of the stimulus or initiating an appropriate response.

Given that cortical malfunctions have been suggested as the cause of sensory differences in autism [15–17] and that the visual cortex has been shown to modulate the response magnitude of looming sensitive cells in the SC [30], we tested whether the behavioural responses are affected in cortex and hippocampus-specific conditional $Setd5$ animals ($Setd5^{+/fl}$; $Emx1$-Cre, S2 Fig). These animals showed no behavioural differences in their reaction times and vigour, or response kinetics (S2C–S2F Fig), in line with previous studies that show that subcortical pathways [29,31], namely, the SC and PAG, and not altered top-down modulation from cortical areas, are required for this behaviour.

## Repetitive looming escape responses

Although the innate LER does not require learning, repeated loom presentations cause adaptive behavioural changes, for example, the emergence of place avoidance of the threat zone [29]. Given that $Setd5^{+/-}$ mice triggered 3 times more looming events and shelter exits than their WT siblings (Fig 1E and 1F), we explored if altered adaptation to repeated presentations could account for the observed difference in average reaction time and vigour (Fig 1G). For

that purpose, we compared the intrinsic behavioural characteristics and the effect of the LER on the exploration strategies and behavioural adaptations across days (Fig 2A–2D). Before stimulus exposure, both cohorts had similar exploratory strategies (S1A–S1G Fig), indicating that their innate exploration strategies, and, thus, intrinsic levels of anxiety, cannot account for the differences in reaction time and vigour. After stimulus exposure, WT animals showed expected reductions in their exploratory behaviour following the initial exposure to the looming stimulus, making fewer exits than during the prestimulus exploration time (Fig 2B), eliciting fewer looms in total (Fig 2C) with relatively constant reaction time (Fig 2D). Strikingly, $Setd5^{+/-}$ consistently triggered more looms and had no signs of sensitisation upon repeated exposures (Fig 2A–2D). After the initial decrease in exploration following the first stimulus presentation, they showed no further consolidation of place avoidance to the threat zone. The reaction time and vigour of the escape response remained longer and slower across days (Fig 2D), respectively, indicating that $Setd5^{+/-}$ do not sensitise as their WT siblings do to the LER paradigm. We next tested if the delayed perceptual decisions (Fig 1) are related to the consolidation of place avoidance, 2 seemingly independent behaviours. We analysed the total number of shelter exits for each animal and compared them with their average reaction times and vigour (Fig 2E and 2F). Surprisingly, the strength of the place avoidance was a strong predictor of the reaction times, but not vigour, suggesting that the timing of the perceptual decision and the formation of place avoidance are intrinsically linked. To assess the limits of response adaptation and to ascertain if the presence of a reward would overcome the fear, we conducted the same behavioural experiment but with the presence of a food reward on the far side of the arena within the threat zone (Fig 2G–2J) and with no interstimulus interval restriction. Although WT siblings did make some attempts to retrieve the food reward, they quickly sensitised, rarely leaving the shelter and were unsuccessful in obtaining the food reward. $Setd5^{+/-}$, on the other hand, increased their shelter-leaving events, often persisting until reaching the reward (Fig 2G–2I and S2 Video). Despite the rapid and repeated exposure, $Setd5^{+/-}$ mice continued to escape to the shelter upon stimulus presentation, for >10 consecutive presentations (Fig 2H–2J). Their reaction times showed a mild adaptation (Fig 2J, top), while vigour remained largely constant (Fig 2J, bottom). These results show that 2 behavioural traits, LER and place avoidance, strongly correlate, suggesting they share a common neuronal pathway.

## Convergent behavioural deficits across ASD models

To test whether the deficiencies observed are specific to $Setd5^{+/-}$ animals or might reveal a more general behavioural phenomenon across ASD models with distinct molecular and genetic aetiologies, we decided to test 2 additional highly penetrant mutations, *Cullin3* (*Cul3*), a ubiquitin ligase-encoding gene, and *Ptchd1* (S3A–S3C Fig), a member of the Patched family speculated to function as a Sonic hedgehog receptor [25]. We exposed sex-matched sibling pairs from these models to the same LER protocol as used for the *Setd5* model and observed qualitatively similar behavioural deficits in reaction time (*Ptchd1*: Fig 3A–3D; *Cul3*: Fig 3I–3L) and place avoidance (*Ptchd1*: Fig 3E–3G; *Cul3*: Fig 3I–3L). Both cohorts required, on average, more repetitions to elicit an escape response (Fig 3A, 3B, 3I, and 3J), had longer reaction times between the stimulus start and the maximum speed (Fig 3C and 3K), displayed slower maximum speed compared to their WT siblings (Fig 3D and 3L), triggering more looms (Fig 3E and 3M) and leaving the shelter more often (Fig 3F and 3N), but had no differences in their behavioural strategies prior to the first loom presentation (Figs 3G–3O, S3D, and S3K). Similar to *Setd5*, these changes were not due to the animals not reacting to the first stimulus, as seen in their consistent arrest behaviour (S3E and S3L Fig), nor to differences in their exploratory behaviour (S3D and S3K Fig). This difference in response reaction time is an intrinsic and not

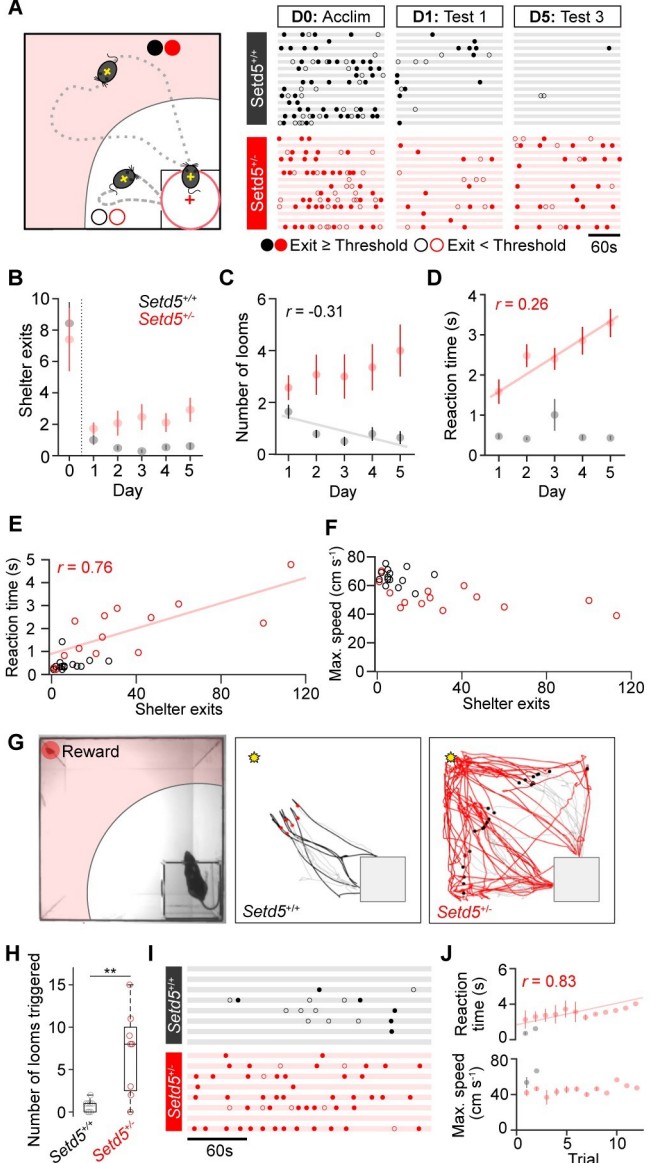

**Fig 2. Altered adaptation and repetitive behavioural phenotype to the LER.** (**A**) Left, graphic depicting exits where the mouse enters (dotted line, filled dot) or does not enter (dashed line, open dot) the threat zone. Right, ethogram of exploratory shelter exit behaviour during the prestimulus acclimatisation, the first and last test of the LER paradigm. Each row represents one animal. (**B-D**) Adaptation in the number of shelter exits, average number of looms triggered and reaction times across days (**B**, Setd5$^{+/+}$: $p = 0.2189$, *Setd5*$^{+/-}$: $p = 0.4974$; **C**, *Setd5*$^{+/+}$, $p = 0.0087$; Setd5$^{+/-}$, $p = 0.2192$; **D**, *Setd5*$^{+/+}$, $p = 0.890$; Setd5$^{+/-}$, $p < 0.001$). (**E**) Relationship between the number of shelter exits and the average reaction time per animal (*Setd5*$^{+/+}$, $p = 0.626$; *Setd5*$^{+/-}$, $p = 0.002$). (**F**) As (**E**) but for maximum escape speed per animal (*Setd5*$^{+/+}$, $p = 0.547$; *Setd5*$^{+/-}$, $p = 0.053$). (**G**) Left, reward trial example showing the location of the food reward within the threat zone. Right, example trajectories during the reward trials show the mouse's position for the 3 s before triggering the loom (light grey) and the 6 s following the stimulus start (black or red). Filled dots represent the position of the mouse when the stimulus was triggered, grey square represents the shelter, and the yellow star shows the position of the food reward. (**H**) Number of looms triggered during the reward trial (*Setd5*$^{+/+}$, 1 bout; *Setd5*$^{+/-}$, 8 bouts, $P = 0.005$). (**I**) Ethograms of exits, as in (**A**), during the reward trials show an increased probability of *Setd5*$^{+/-}$ mice leaving the shelter during a trial (number of exits, 2.12 for *Setd5*$^{+/+}$, 6.88 for *Setd5*$^{+/-}$, $p = 0.057$). (**J**) Reaction time (top panel, *Setd5*$^{+/-}$, 44 trials, $r = 0.828$, $p = 0.0001$) and escape vigour (bottom panel, *Setd5*$^{+/-}$, 44 trials, $r = 0.184$, $p = 0.5116$) during repeated presentations of the loom. Trials when the animal was interacting with the reward were excluded. *P*-values: Wilcoxon's test, *p*-values: Pearson's correlation test, unless specified. Plotted linear fits depict the statistically significant correlations. The data underlying this figure can be found in S2 Data.

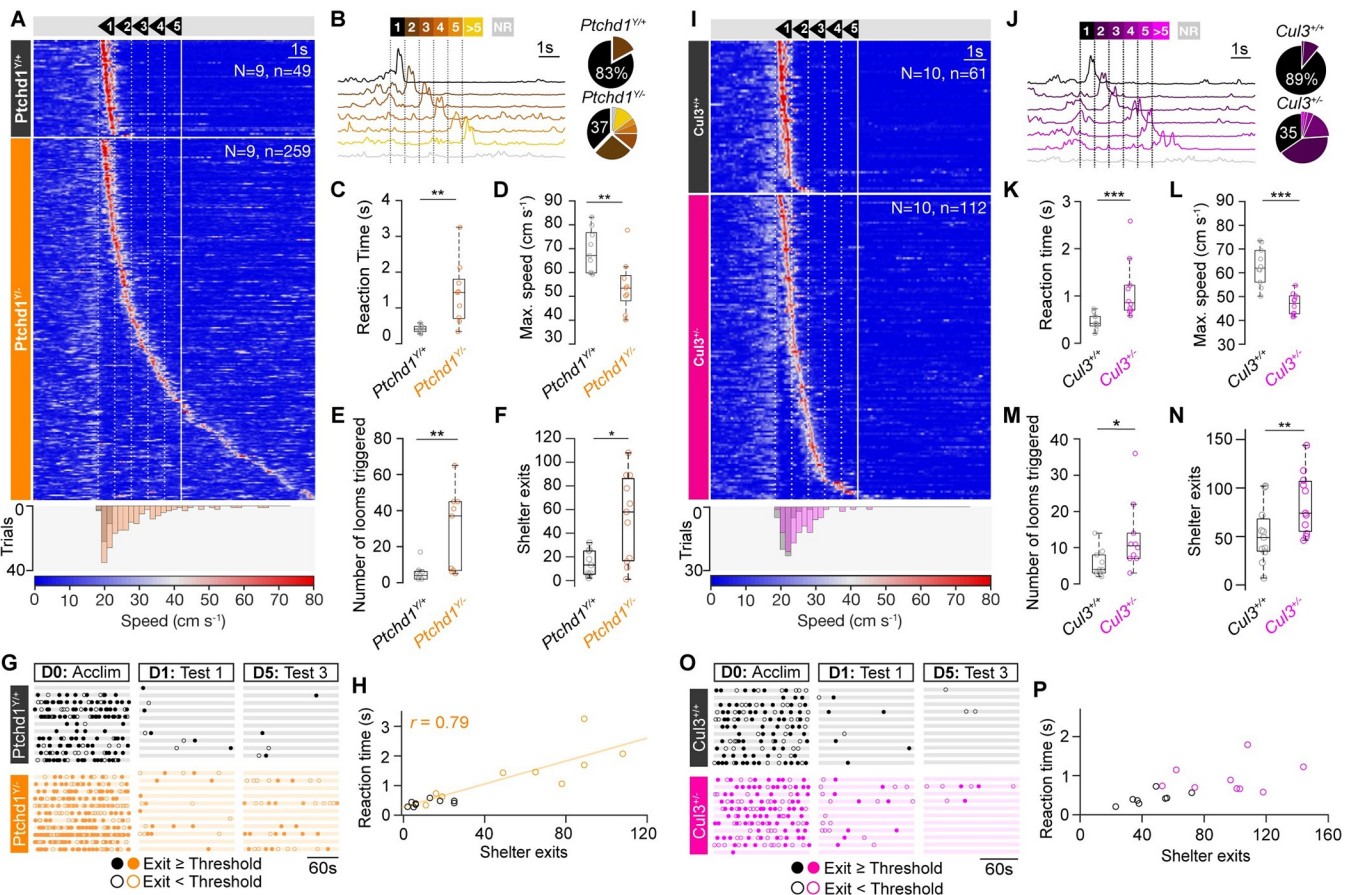

**Fig 3. Conserved behavioural changes across etiologically distinct autism models.** (**A**) Raster plot of mouse speed in response to the looming stimuli for *Ptchd1*$^{Y/+}$ (upper, *n* = 9, 49 trials) and *Ptchd1*$^{Y/−}$ (lower, *n* = 9, 259 trials), sorted by reaction time. Bottom, distribution of reaction times for all *Ptchd1*$^{Y/+}$ (black) and *Ptchd1*$^{Y/−}$ (orange) trials. (**B**) Left, example trials based on whether the mouse responds within 1 of the 5-loom stimuli, after the fifth (>5), or not at all (NR, no response) from 1 *Ptchd1*$^{Y/−}$ mouse. Right, proportion of escape to loom presentations. Trials where mice escaped within the first loom: *Ptchd1*$^{Y/+}$, 0.826, *Ptchd1*$^{Y/−}$, 0.374, *p* = 0.004. (**C**) Average reaction time and (**D**) maximum escape speed per animal, for all trials where the mice escaped (reaction time; *Ptchd1*$^{Y/+}$, 0.411 s, *Ptchd1*$^{Y/−}$, 1.41 s, *p* = 0.002; maximum escape speed; *Ptchd1*$^{Y/+}$, 69.1 cm s$^{−1}$, *Ptchd1*$^{Y/−}$, 54.2 cm s$^{−1}$, *p* = 0.005). (**E**) Average total looms triggered per genotype across 5 days of testing (*Ptchd1*$^{Y/+}$, 5.44 looms; *Ptchd1*$^{Y/−}$, 28.8 looms, *P* = 0.004). (**F**) Average total shelter exits across the 5 test days (*Ptchd1*$^{Y/+}$, 15; *Ptchd1*$^{Y/−}$, 53, *P* = 0.025). (**G**) Ethogram of shelter exits during the prestimulus acclimatisation, first and last trial of the LER paradigm. Each row represents one animal, filled and open dots represent exits that crossed into the threat zone or not, respectively. (**H**) Relationship between the number of shelter exits and the average reaction time per animal (*Ptchd1*$^{Y/+}$, *p* = 0.126; *Ptchd1*$^{Y/−}$, *r* = 0.76, *p* = 0.012, Pearson's correlation). (**I-P**) Same as (**A-H**) but for *Cul3*. (**I**) *Cul3*$^{+/+}$ (top, *n* = 10, 61 trials) and *Cul3*$^{+/−}$ (bottom, *n* = 10, 112 trials). Bottom, distribution of reaction times (*p* < 0.001, two-way KS test). (**J**) *Cul3*$^{+/+}$, 0.892, *Cul3*$^{+/−}$, 0.347, *p* < 0.001, two-way KS test). (**K**) Reaction time; *Cul3*$^{+/+}$, 0.464 s, *Cul3*$^{+/−}$, 1.11 s, *P* < 0.001. (**I**) Maximum escape speed; *Cul3*$^{+/+}$, 62.7 cm s$^{−1}$, *Cul3*$^{+/−}$, 46.8 cm s$^{−1}$, *P* < 0.001). (**M**) (*Cul3*$^{+/+}$, 5.60 looms; *Cul3*$^{+/−}$, 12.9 looms, *P* = 0.023). (**N**) (*Cul3*$^{+/+}$, 47; *Cul3*$^{+/−}$, 89, *P* = 0.006). (**P**) (*Cul3*$^{+/+}$, *p* = 0.078; *Cul3*$^{+/−}$, *p* = 0.571, Pearson's correlation). Box-and-whisker plots show median, IQR, and range. Shaded areas represent SEM. Lines are shaded areas, mean ± SEM, respectively. *P*-values are Wilcoxon's test, *p*-values: two-sample Kolmogorov–Smirnov test, unless specified. The data underlying this figure can be found in S3 Data.

an adaptive property since it is present during the first loom presentation (S3G and S3N Fig). However, they varied in strength, with *Cul3*$^{+/−}$ mice showing a less severe phenotype than the *Setd5*$^{+/−}$ or *Ptchd1*$^{Y/−}$ models. Interestingly, slower reaction times and reduced vigour could also be observed in *Ptchd*$^{Y/−}$ animals, despite their baseline hyperactivity (S3D and S3I Fig). Interestingly, in both cohorts, the strength of the place avoidance and the reaction times were significantly lower and longer, respectively (Fig 3C, 3F, 3K and 3N), being linearly correlated in *Ptchd1*$^{Y/−}$, but not *Cul3*$^{+/−}$ (Fig 3H and 3P). These results suggest a general link between the timing of the perceptual decision and the formation of place avoidance, despite different molecular and developmental origins.

## Deep medial SC, not retinal drive, is linked to delay in LER

Changes in average reaction times have previously been shown to depend on the saliency of the threat stimulus. Whereas repeated high-contrast looming stimuli evoke strong LER, low-contrast looms drive less vigorous escapes and longer reaction times [29], suggesting that deficits in visual processing could underlie the behavioural differences observed in $Setd5^{+/-}$. This possibility is aggravated by the fact that the expression of genes involved in eye development are known to be misregulated in $Setd5^{+/-}$ animals [23]. To explore possible visual response changes, we recorded visually evoked responses across different layers of the SC, using 32-channel silicon probes in head-fixed, awake-behaving animals (S4A Fig), and determined the recording depth using current source density analysis (S4B and S4C Fig). In our setup, mutant and control animals had similar running probabilities and showed similar baseline firing properties (S4D Fig). We next assessed visual response properties to full-field flashes, visual stimuli that drive all neurons similarly, irrespective of their receptive field positions. First, we show that pupil dynamics are unaffected across genotypes (S4E Fig). Next, when clustering visually responsive cells into 10 clearly defined groups, we could observe similar proportions and identical response properties (S4F and S4G Fig). Identical responses were also observed in their spatiotemporal receptive fields and the proportion of ON and OFF-responsive cells (S4A–S4G Fig). Direction-selective responses (S4H and S4I Fig) were also indistinguishable across groups. Finally, we explored the response properties to looming stimuli across SC layers. Looms elicit robust and reproducible firing across genotypes (S4J Fig), with similar firing, peak rates, and time-to-peak responses across genotypes (S4K–S4M Fig). The overall response kinetics (S4N Fig) and the adaptation in average firing rates across presentations (S4O and S4P Fig) were indistinguishable across genotypes. Similarly, no differences were observed when comparing the visual response properties of Cul3 and Ptchd1 models (S5 Fig). The lack of any reportable visual response difference across the SC layers indicates that retino-collicular visual processing is largely unaltered and excludes the possibility that sensory processing can account for the observed visuomotor deficits and suggests that the main cause of the behavioural difference may reside in downstream areas. Given that optogenetic stimulation of deep medial SC (dmSC) neurons that project to the dPAG generates robust escape responses and that the dPAG encodes for the choice of escape and controls the escape vigour [29], we hypothesised that activating dmSC neurons would reveal behavioural differences between genotypes and delineate the underlying circuit impairments. To test this, we focused on the $Setd5$ mouse line and used unilateral in vivo channelrhodopsin-2 (ChR2) activation of VGluT2$^+$ cells in the dmSC (Fig 4A–4C). In line with previous findings, activation of dmSC neurons in $Setd5^{+/+}$ animals instructed immediate shelter-directed escapes (Fig 4D) and were absent in controls (S6 Fig). Gradually increasing the laser power and, thus, optogenetic activation, led to corresponding decreases in the latency to respond and an increase in the escape vigour in $Setd5^{+/+}$ (Fig 4C). By contrast, activation of dmSC neurons in the $Setd5^{+/-}$ background more often resulted in an initial arrest, followed by shelter-seeking escape responses (Fig 4F and 4G), mimicking the behavioural divergence observed for LER (Fig 1). Interestingly, this behavioural difference was evident only during strong optogenetic activations. At lower laser intensities, both genotypes exhibited a transition from arrest to weak escape behaviours (Fig 4H) with similarly low reaction time (Fig 4E and 4G) and vigour (Fig 4H). Correspondingly, the escape probabilities were identical at low optogenetic stimulation but diverged with increased laser power (Fig 4I). Given that changes in the strength of optogenetic activation mimic LER to looms of different contrast [29], we hypothesised that when probing animals to LER of different contrasts, both genotypes should show similar responses at lower contrast (Fig 4J). To test this, we performed the LER at 3 different contrast levels: 98%, 50%,

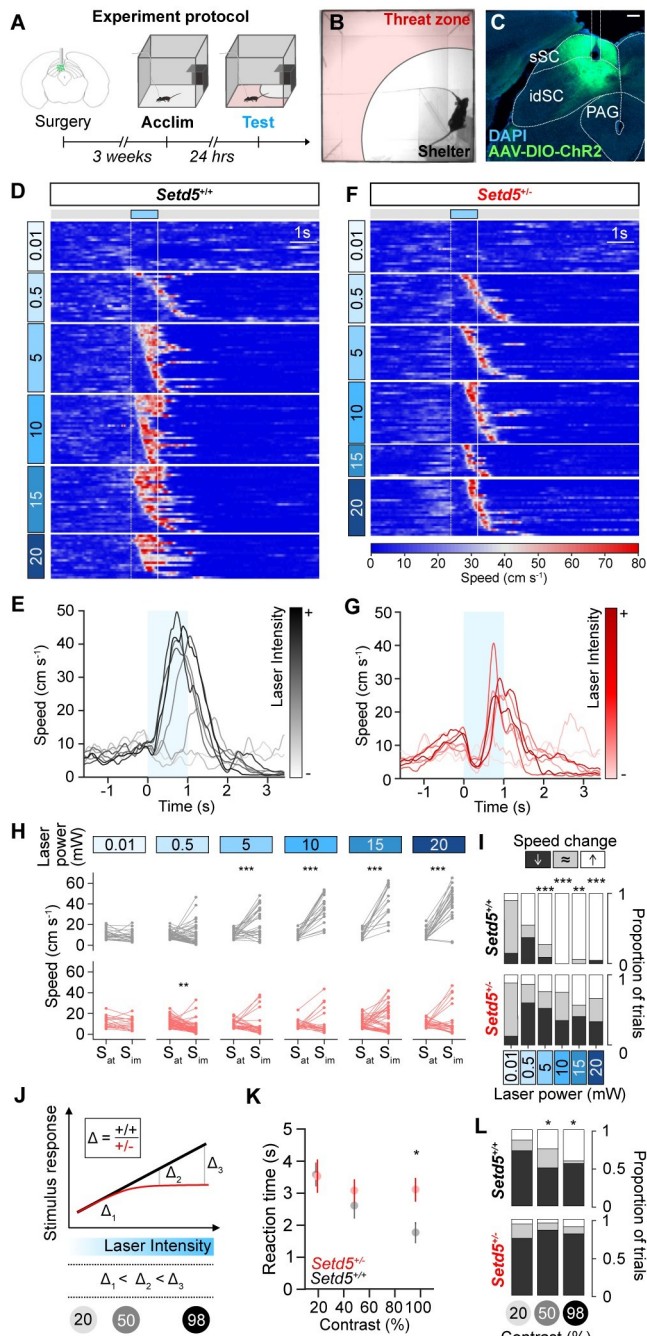

**Fig 4. Activation of deep SC neurons recapitulates delayed LER in *Setd5*$^{+/-}$ animals.** (**A**) Timeline of the experimental protocol for optogenetic activation of the dmSC. (**B**) Video frame during an optogenetics trial. (**C**) Confocal micrograph of AAV-ChR2 expression in dmSC and optic fibre location reconstruction, scale bar: 200 μm. (**D**) Raster plot of mouse speed in response to optogenetic activation, sorted by laser intensity and (**E**) mean speed responses at increasing laser intensities for *Setd5*$^{+/+}$ (*n* = 4, 312 trials). (**F, G**) As (**D, E**) but for *Setd5*$^{+/-}$ mice (*n* = 4, 291 trials). Blue-shaded areas represent the laser stimulation. (**H**) Subplots of trials in (**D, F**) showing the immediate change in speed upon light activation at different laser intensities for *Setd5*$^{+/+}$ (*n* = 4, paired *t* tests) and *Setd5*$^{+/-}$ (*n* = 4, paired Wilcoxon's tests) optogenetics trials. $S_{at}$ is the mean speed of the animal ±50 ms of laser onset, and $S_{im}$ is the mean speed of the animal 300–800 ms after laser onset. (**I**) Proportion of trials at different laser intensities that either show an increase (white), decrease (black), or no change (grey) in speed upon light activation for *Setd5*$^{+/+}$ (top) and *Setd5*$^{+/-}$ (bottom). 0.01 mW mm$^{-2}$: 35 trials, *p* = 0.649; 0.5 mW mm$^{-2}$: 111 trials, *p* = 0.193; 5 mW mm$^{-2}$: 50 trials, *p* < 0.001; 10 mW mm$^{-2}$: 49 trials, *p* < 0.001; 15 mW mm$^{-2}$: 61 trials, *p* = 0.004; 20 mW mm$^{-2}$: 49 trials, *p* < 0.001, X$^2$ test of independence. (**J**) Schematic of the behavioural divergence between genotypes with increasing stimulus

intensity (laser or loom). (**K**) Summary of mean ± SEM of reaction time to the LER paradigm at different stimulus contrasts ($p = 0.021$ for the interaction between genotype and contrast, $Setd5^{+/+}$, $n = 13$, 68 trials; $Setd5^{+/-}$, $n = 13$, 59 trials, repeated measures ANOVA. $p = 0.018$ for 98% contrast, with multiple comparisons and Bonferroni correction. $Setd5^{+/+}$, 24 trials; $Setd5^{+/-}$, 22 trials). (**I**) Proportion of trials at different contrast looms that show an increase (white), decrease (black), or no change (grey) in speed upon light activation (20%: $p = 0.698$; 50%, $p = 0.026$; 98%, $p = 0.021$, $X^2$ test of independence). The data underlying this figure can be found in S4 Data.

and 20% (Fig 4K and 4L). As predicted, the LER to low-contrast looms did not differ across genotypes, showing similar reaction times (Fig 4K). Remarkably, in $Setd5^{+/-}$ animals, the reaction time remained largely constant across contrast levels (Fig 4F and 4G), suggesting that the time required to initiate a response is independent of stimulus intensity. Together with the persistence of the arrest behaviour even at high levels of optogenetic activation (Fig 4H), this supports the idea that there is a functional bottleneck downstream of the sensory processing circuits within the SC that impairs the timely generation of escape responses.

## dPAG neurons are hypoexcitable in the *Setd5* mouse model

Since our previous results point towards a disruption in the circuits downstream of the SC, we decided to investigate if the activity of the underlying dPAG is affected in the $Setd5^{+/-}$ model (Fig 5). To test if any synaptic or intrinsic properties of dPAG neurons are impaired, we performed whole-cell patch-clamp recordings in slices (Fig 5A). First, using the same animals used for optogenetic stimulation in vivo, we tested if the synaptic properties of dmSC inputs to the dPAG are changed. Ex vivo optogenetic activation elicited monosynaptic excitatory postsynaptic current (EPSC) with an average current of −244 ± 9.65 pA in $Setd5^{+/+}$ and −228 ± 11.8 pA in $Setd5^{+/-}$ (Fig 5A and 5B). At this frequency, repeated stimulations showed no facilitation in both genotypes (Fig 5B), indicating that dmSC inputs to the dPAG are unaltered. Next, we explored spontaneous neurotransmission by recording sEPSCs (S7A–S7C Fig), supporting the view that the synaptic properties remain similar at a circuit level. Finally, we probed the intrinsic properties of dPAG neurons. These experiments were done blind to the cell type, but we were able to classify them based on their firing statistics as excitatory and inhibitory [32] (S7F–S7J Fig). Although the membrane potential, input resistance, membrane constant, and capacitance were identical between genotypes (Fig 5C), we observed a stark decrease in the ability of these cells to generate action potentials in response to injected current in both excitatory and inhibitory neurons (Figs 5D and S7H), without affecting the rheobase (S7K Fig). These changes were accompanied by changes in the spike shape of $Setd5^{+/-}$ animals (Fig 5E and 5F) that showed slightly higher depolarisation, lower after-hyperpolarisation and slower rise dynamics in comparison to their WT siblings (S7I and S7J Fig). The hypoexcitability phenotype is also pronounced when the same analysis is performed on dPAG cells, which have been shown to receive input from dmSCs (S7K Fig) using optogenetic activation of dmSC fibres (Fig 5A and 5B). Based on prior work, we know that stronger dPAG activation leads to more vigorous escapes and that the underlying biophysical mechanisms depend on dPAG neuronal synaptic integration and excitability [29]. Thus, intrinsic excitability changes in excitatory neurons are a proxy of the expected escape strength. Remarkably, the response characteristics of dPAG neurons in $Setd5^{+/+}$ and $Setd5^{+/-}$ match their behavioural differences (Fig 1). At weak current injections (Fig 5D), optogenetic activation (Fig 4D–4H), or low contrast looms (Fig 4K), both genotypes are indistinguishable from each other. However, at stronger current injections $Setd5^{+/+}$ and $Setd5^{+/-}$ diverge, with a hypoexcitability phenotype for $Setd5^{+/-}$ (Fig 5D) that matches the delay in LER for optogenetic (Fig 4D and 4F) and visual (Figs 1 and 4K) activations. To determine if the hypoexcitability phenotype was present in the preceding visual

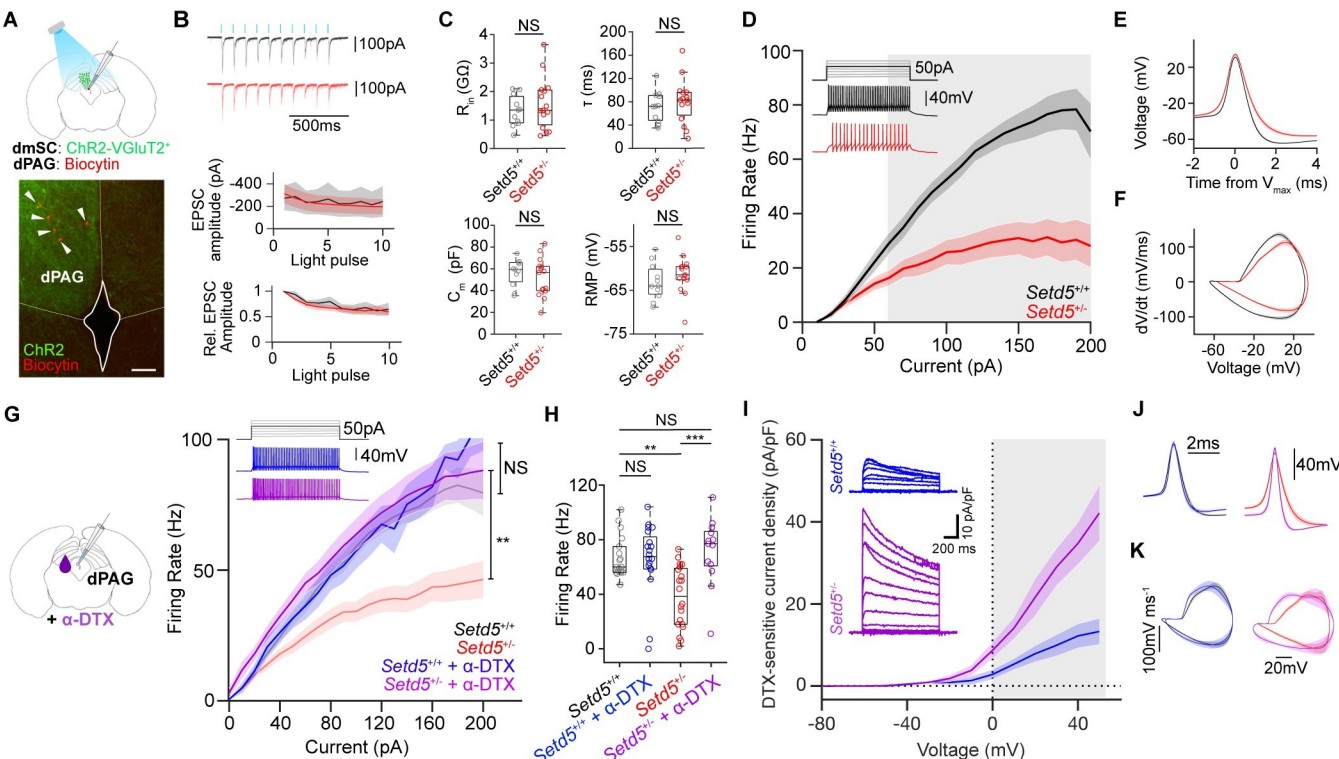

**Fig 5. *Setd5^{+/−}* dPAG cells are hypoexcitable.** (**A**) Schematic of the experimental approach, in vitro patch clamp recordings (top) and micrograph of VGluT2^+ dmSC projections to the dPAG infected with AAV9-ires-DIO-ChR2 (green) with dPAG biocytin filled and recorded cells (red and arrow heads, scale bar: 100 μm). (**B**) Top, whole-cell voltage clamp traces of example *Setd5^{+/+}* and *Setd5^{+/−}* cells (black and red, respectively) responding to 10 Hz light stimulation (blue ticks). Amplitude (middle, *p* = 0.078) and relative EPSC amplitude (bottom, *p* = 0.565) of responses to sequential light pulses in a 10-Hz train. (**C**) Intrinsic properties of *Setd5^{+/+}* (*n* = 6, 11 cells) and *Setd5^{+/−}* (*n* = 7, 14 cells) dPAG cells. Input resistance (*P* = 0.756, top left), membrane constant tau (*P* = 0.436, top right), membrane capacitance (*P* > 0.995, bottom left) and resting membrane potential (*P* = 0.213, bottom right). (**D**) Summary of the relationship between current injection and action potential firing for putative glutamatergic cells showing a strong reduction in firing (*p* < 0.001). Inset, representative example traces to a 50-pA current injection. Grey area indicates the current injection values that significantly differ between *Setd5^{+/+}* cells (black) and *Setd5^{+/−}* cells (red) found by a multiple comparisons analysis with Tukey correction. (**E**) Average shape and (**F**) phase plane analysis of the action potentials generated in the rheobase sweep (*Setd5^{+/+}*, 13 cells, 42 spikes; *Setd5^{+/−}*, 14 cells, 72 spikes). (**G**) Summary of the relationship between current injection and action potential firing for all *Setd5^{+/+}* cells (black, *n* = 18) and *Setd5^{+/−}* cells (red, *n* = 20) before and after (*Setd5^{+/+}*: blue, *n* = 12; *Setd5^{+/−}*: purple, *n* = 13 cells) application of α-Dendrotoxin (α-DTX, 100 nM, *p* > 0.995 and *p* = 0.0147 for the effect of α-DTX on *Setd5^{+/+}* and *Setd5^{+/−}* firing, respectively). Inset, representative example traces from *Setd5^{+/+}* cells (blue) and *Setd5^{+/−}* (purple) cells after α-DTX application to a 50-pA current injection. (**H**) Effect of α-DTX on firing in response to 120 pA current injection. Before α-DTX (*Setd5^{+/+}* versus *Setd5^{+/−}*; *p* = 0.0017), effect of α-DTX on *Setd5^{+/+}* (before versus after α-DTX; *P* > 0.995) and *Setd5^{+/−}* (before versus after α-DTX; *P* < 0.001, *Setd5^{+/+}* before versus *Setd5^{+/−}* after α-DTX; *P* > 0.995). Multiple comparison analysis after rm-ANOVA with Tukey correction. (**I**) α-DTX-sensitive current densities in *Setd5^{+/+}* and *Setd5^{+/−}* dPAG neurons (*p* < 0.001). Inset, example of α-DTX-sensitive traces (*Setd5^{+/+}*: black, *n* = 7 cells; *Setd5^{+/−}*: red, *n* = 6 cells). Grey area indicates the current values that significantly differ between Setd5^{+/+} cells (black) and Setd5^{+/−} cells (red) found by a multiple comparisons analysis with Tukey correction. (**J**) Action potential shape and (**K**) phase plane analysis of the action potentials generated in the rheobase current in *Setd5^{+/+}* cells (without α-DTX, black, *n* = 19; with α-DTX, blue, *n* = 12), *Setd5^{+/−}* cells (without α-DTX, red, *n* = 21; with αDTX, purple, *n* = 13). The data underlying this figure can be found in S5 Data.

pathway, we patched dmSC interneurons (S7L–S7R Fig). Here, no statistical significance was observed despite a hypoexcitability trend, indicating that the strength of the effect is area-specific but not ruling out other brain-wide deficiencies. In combination with previous work [29] and our optogenetic experiments (Fig 4), these results suggest intrinsic deficits in the dPAG account, at least partly, for the delayed LER phenotype in the *Setd5^{+/−}* mice (Fig 1).

## dPAG cells are hypoexcitable due to $K_v$ channel misregulation

Next, we investigated the underlying mechanisms of the hypoexcitable phenotype of the *Setd5^{+/−}* dPAG cells. The changes in excitability and spike kinetics are indicative of differences in channel composition that are part of adaptive homeostatic mechanisms [33], suggesting the

 

influence of voltage-gated potassium channels (Kv). Previously published work [23], which studied the transcriptomic changes brought about by a mutation in $Setd5^{+/-}$, highlighted significant up-regulation in the expression of Kv1.1 channels during development. To test whether the blockade of Kv1.1 would rescue the hypoexcitability phenotype, we applied 100 nM of **α**-Dendrotoxin (**α**-DTX), a specific Kv1.1, Kv1.2, and Kv1.6 blocker, and patched dPAG cells. While **α**-DTX had no effect on $Setd5^{+/+}$ excitability of dPAG cells, strikingly, $Setd5^{+/-}$ cells reversed their hypoexcitability phenotype to WT levels (Fig 5G and 5H). Accordingly, we measured a drastically increased **α**-DTX sensitive outward current in Setd5$^{+/-}$ dPAG cells (Fig 5I). In addition, after **α**-DTX application, the spike dynamics remained identical in $Setd5^{+/+}$, whereas $Setd5^{+/-}$ became faster, mimicking the WT properties (Fig 5J and 5K). These differences are not due to an overexpression of Kv channels. Tissue-specific proteomic analysis showed no difference in the protein level of Kv channels in either the PAG (Fig 6A), SC, or cortex (S8A and S8B Fig). This is in accordance with western blot and immunohistochemical analysis (Figs 6B, 6D, S8C and S8D). Given that Kv1.1 has been recently shown to be involved in the homeostatic control of firing [33,34] through modulation of the axon initial segment (AIS) [35], these results indicate that the deficits might not arise due to direct changes in expression but due to homeostatic misregulation. Next, we tested if the dPAG neurons of $Ptchd1^{Y/-}$ and $Cul3^{+/-}$ animals showed similar physiological properties to $Setd5^{+/-}$ neurons. As expected from the molecular differences among the tested ASD models, the physiological characteristics differed. *Ptchd1* animals, despite having mostly identical intrinsic properties to their WT siblings (S9A–S9D Fig), also show a hypoexcitability phenotype at high current inputs (S9E–S9I and S9S Fig). However, this hypoexcitability phenotype cannot be rescued by **α**-DTX application (S9S Fig). *Cul3* animals did not show any changes in either their current-firing relationship or intrinsic properties (S9J–S9R, S9U and S9V Fig). Overall, these results indicate that different models might have distinct neurodevelopmental origins or key molecular changes that give rise to particular behavioural traits. In particular, for *Setd5*, Kv1 channels appear as key players and targets to reverse the delayed LER phenotype and probe the causality of its correlation with place avoidance deficits.

To investigate whether the area-specific dPAG hypoexcitability phenotype mediated the delayed LER, we tested whether **α**-DTX application targeted directly to the dPAG could rescue the behavioural phenotype (Fig 6A). We implanted cannulas above the dPAG in $Setd5^{+/-}$ and their WT littermates, a target-specific approach as visualised through neurobiotin injections (Fig 6B). Initially, we injected saline as a control in both cohorts and subsequently tested the LER (Fig 6C and 6D, top). As previously demonstrated (Fig 1), $Setd5^{+/-}$ animals responded slower than their WT littermates and had no difference in their maximal escape speed during the first presentation day (Fig 6D, top). However, the behaviour of both cohorts were indistinguishable from each other when **α**-DTX was applied, escaping immediately after the loom presentation (Fig 6C and 6D, bottom, and S3 Video), with similar reaction times (Fig 6E) and vigour (Fig 6F), even despite $Setd5^{+/-}$ animals normally adapting across days (Fig 2D). We then quantified the shelter exits of both cohorts (Fig 7G and 7H) to investigate whether the correlation between shelter exits and delayed LER (Fig 2C) could have a common origin. Remarkably, $Setd5^{+/-}$ animals recovered their place avoidance as well after **α**-DTX application (Fig 6H), indicating that the timely initiation of actions and memory formation are closely linked. These results suggest that perceptual deficits can have far-reaching consequences for unrelated behavioural traits.

## Discussion

ASDs are widespread in the global population, but our understanding of the disorder's origins remains limited. Advancements in genetic sequencing technologies have enabled the isolation of autism-risk genes, opening the door to several in-depth studies into the molecular

 

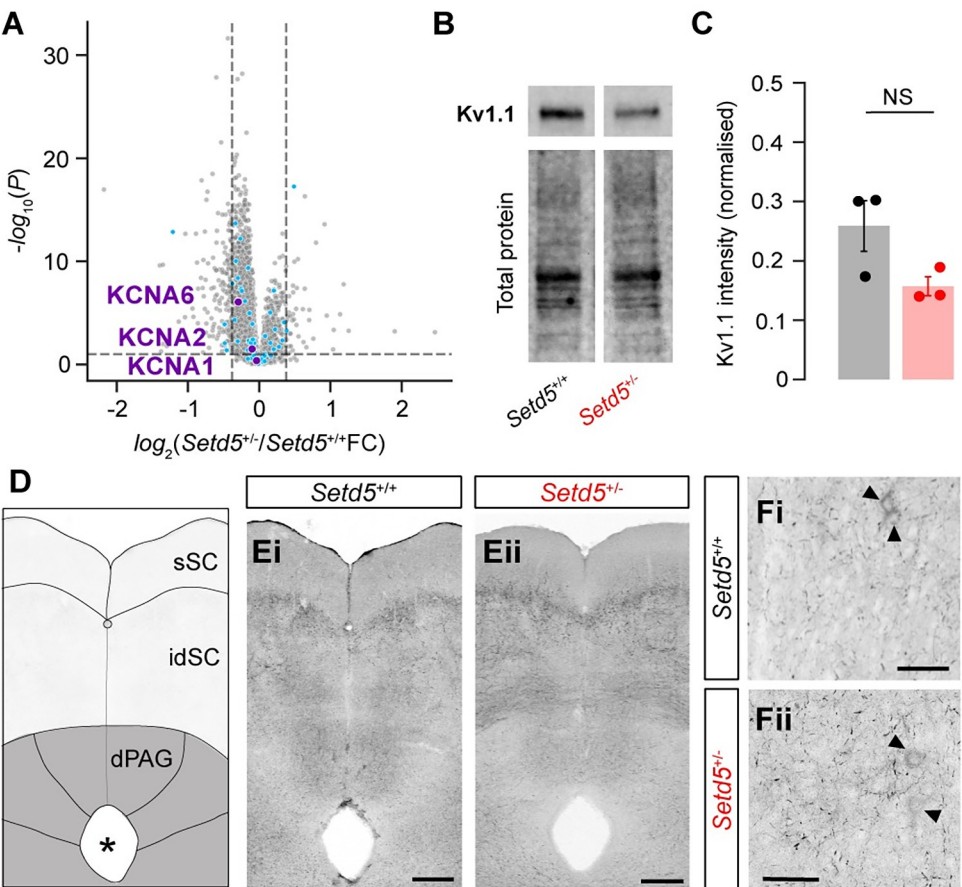

**Fig 6. Protein levels of Kv channels are not changed in *Setd5*⁺ᐟ⁻ dPAG cells.** (**A**) Volcano plot for differential protein levels between adult S*etd5*⁺ᐟ⁺ and *Setd5*⁺ᐟ⁻ mice (*n* = 6 samples per genotype, cyan dots represent proteins annotated as ion-channels and purple dots represent Kv1.1, Kv1.2, and Kv1.6. Lines and shaded areas, mean ± SEM, respectively. Box-and-whisker plots show the median, IQR, and range. *P*-values are Wilcoxon's rank sum test. *p*-values are two-way repeated measures ANOVA. Volcano plot horizontal dashed line represents the significance threshold (*P*-value < 0.1) from a two-sided moderated *t* test, while the vertical dashed lines indicate fold change values greater or lower than 0.4 between *Setd5*⁺ᐟ⁺ and *Setd5*⁺ᐟ⁻. (**B, C**) Tissue-specific western blots of Kv1.1 protein content in the dorsal periaqueductal grey (dPAG) for *Setd5*⁺ᐟ⁺ and *Setd5*⁺ᐟ⁻ mice and (**C**) their quantification (dPAG: *Setd5*⁺ᐟ⁺, 0.259; *Setd5*⁺ᐟ⁻, 0.157, *P* = 0.090). (**D**) Schematic of SC and PAG regions of interest. (**E, F**) Antibody staining for Kv1.1 in (**E**) Setd5⁺ᐟ⁺ and (**F**) Setd5⁺ᐟ⁻ (**E**, 55 μm projection; **F**, 30 μm projection). Arrowheads indicate somas stained for Kv1.1. Scale bar: **D**: 200 μm; **E**: 50 μm; **F**: 100 μm. *P*-values are two-tailed Wilcoxon's signed-rank test. The data underlying this figure can be found in S6 and S7 Data.

mechanism of action of these genes [36]. However, there is a considerable mismatch between the observed wealth of molecular changes and our understanding of their roles in changing circuit function and, correspondingly, behaviour. The latter is particularly relevant since ASD is diagnosed by a combination of behavioural traits known as diagnostic criteria [2]. Although several behavioural paradigms are being studied across mouse models [37], the variability and experimental intricacies of behavioural studies make direct comparisons across models difficult. This has led to studies focusing on single models, defining particular changes but not general principles [38]. Furthermore, these studies have primarily centred on the intricacies of social interactions, communication, and repetitive behaviours—complex traits that present a formidable challenge in establishing a clear connection between neuronal dysfunction and behavioural manifestations. Here, we show that the study of innate defensive behaviours, a

perceptual task, provides a convergent behavioural framework that enables the directed analysis of the underlying sensorimotor processes up to the molecular level, permitting comparative dissections of the neuronal dysfunctions across ASD models and a systematic understanding of the neuronal mechanisms underlying the co-occurrence of behavioural traits. This is particularly interesting as the severity of sensory and perceptual impairments has been strongly linked to the strength and expression of traditional, nonsensory ASD traits [18–20].

Defensive escape behaviours are among the most fundamental perceptual decisions performed by animals [22]. Their finely-tuned mechanisms are indispensable for survival, but also for properly interacting with the environment. Whereas some stimuli unambiguously signal an imminent threat and should instruct immediate action, others are ambiguous and require adaptive responses arbitrated by the current context and state to decide, e.g., between ignoring, freezing, fighting, or escaping [22]. In mice, behavioural escape decisions are known to be initiated by the dPAG [29], where appropriate escape decisions are thought to be determined. While all tested ASD models can, in principle, respond behaviourally as robustly as their WT siblings, they require longer and respond with less vigour once an action has been initiated, hampering their ability to develop an appropriate place avoidance to the threat zone (Figs 1–3). The underlying changes were rigorously dissected in the *Setd5* haploinsufficient model, pointing to a specific misregulation of voltage-gated potassium channels in dPAG neurons that gives rise to a strong hypoexcitability phenotype (Figs 3–5), namely, Kv1.1, Kv1.2, and Kv1.6. Interestingly, it is not the level of expression of the channel that appears to be important, as no difference was found between Setd5+/− and Setd5+/+ at the protein level (Fig 6). This indicates that Kv1 channels may be generally inactive in WT animals, as evidenced by the lack of increase in excitability in Setd5+/+ animals during α-DTX application, and suggests that changing the conductance of Kv channels may be a mechanism for behavioural adaptation to threatening stimuli, such as imminent suppression of escape [39]. Targeted pharmacological rescue of this hypoexcitability in vivo completely reverses the behavioural phenotype (Fig 7). This is an important finding since other brain areas are also known to be required for proper defensive responses, in particular, freezing, such as the pathway to the basolateral amygdala (BLA) via the lateral posterior thalamus (LP) [9,40]. Our results indicate a specific involvement of the dPAG and not the BLA via LP pathway.

In addition to the delayed LER, we observed a strong reduction in place avoidance to the threat zone that elicited maladapted repetitive behaviour (Fig 2 and S2 Video). This suggests that ASD mouse models either have deficits in fear memory formation or are intrinsically less fearful, thus incapable of appropriately interpreting the noxiousness of the threatening experience (Figs 1 and 2 and S1 and S2 Videos)—maladaptive behaviours that recapitulate some human ASD traits [41]. Interestingly, this reduction of place avoidance is directly linked with longer reaction times to the LER across ASD models (Figs 2E, 3H and 3P), and, thus, with the dPAG hypoexcitability phenotype (Fig 5D). We show that this correlation is causally linked, given that the target-specific pharmacological rescue reverses both, the delayed LER and place avoidance to WT levels (Fig 7 and S3 Video). Thus, maladaptive perceptual decisions, as with the delayed LER, can profoundly affect seemingly independent behavioural traits, such as LER, threat-induced place avoidance and the emergence of repetitive behaviours (S1 and S2 Videos). This is in line with studies in humans. Threat imminence has been shown to elicit PAG activation [42] and the electrical stimulation of midbrain structures elicited strong emotional reactions [43]. In rodents, the dPAG has been shown to support fear learning [44], particularly in contextual conditioning paradigms [45]. Overall, our results emphasise the role of subcortical pathways through the PAG in the altered perceptual abilities frequently described in ASD, and fear memory formation in general.

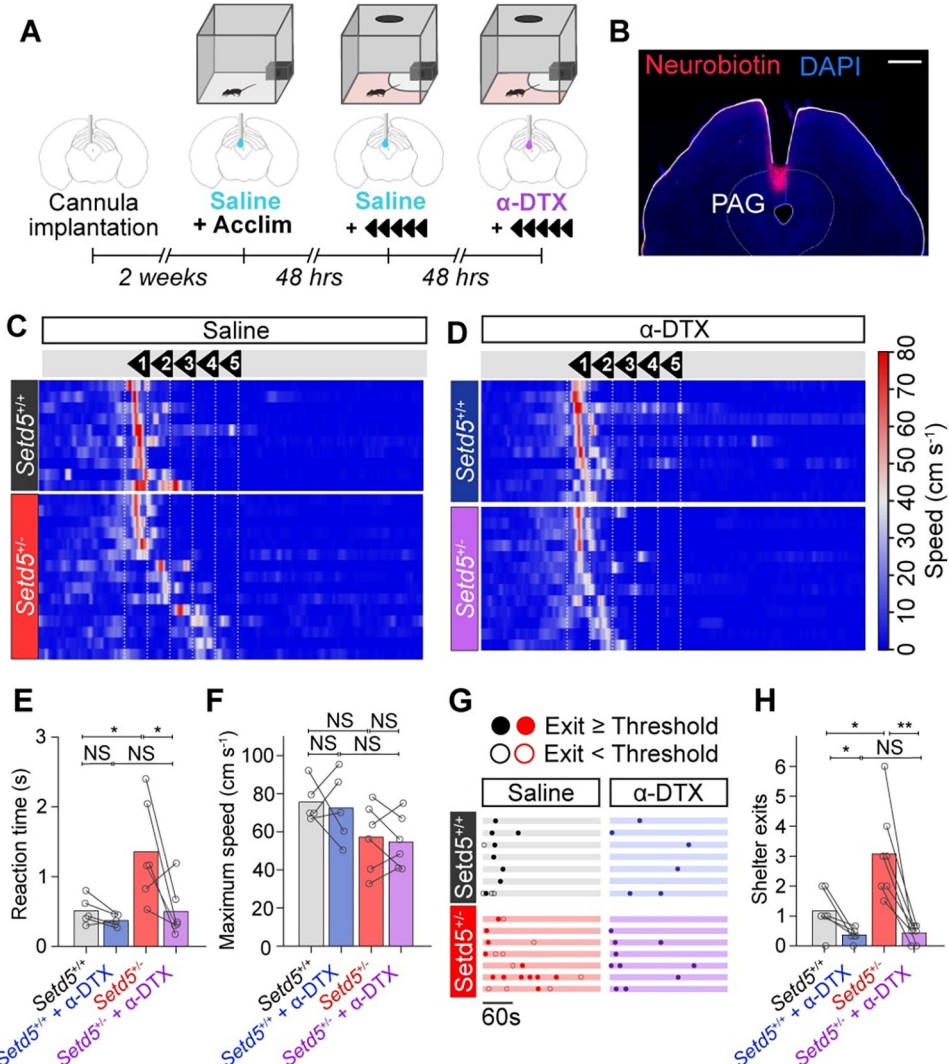

**Fig 7. Pharmacological Kv channel block rescues delayed LER and place avoidance. (A)** Timeline of in vivo **α**-DTX cannula experiments. (**B**) Confocal micrograph of a coronal section of the location of the cannula above the dPAG as well as the expression of neurobiotin that was infused into the cannula at the end of the experiments. Scale bar: 500 μm. (**C**) Raster plots of mouse speed in response to the looming stimuli for *Setd5*⁺/⁺ before (top, *n* = 6, 10 trials) and after (bottom, *n* = 6, 11 trials) infusion of **α**-DTX (500 nL, 500 nM) sorted by reaction time. (**D**) As for (**C**) but for *Setd5*⁺/⁻ before (top, *n* = 6, 15 trials) and after (bottom, *n* = 6, 13 trials) animals after infusion of **α**-DTX (500 nL, 500 nM). (**E**) Effect of **α**-DTX on reaction time. Before **α**-DTX (*Setd5*⁺/⁺ versus *Setd5*⁺/⁻; *P* = 0.017), effect of **α**-DTX on *Setd5*⁺/⁺ (saline versus **α**-DTX; *p* = 0.104) and *Setd5*⁺/⁻ (saline versus **α**-DTX; *p* = 0.014) and after **α**-DTX (*Setd5*⁺/⁺ versus *Setd5*⁺/⁻; *P* = 0.571). (**F**) Effect of **α**-DTX on escape vigour. Before **α**-DTX (*Setd5*⁺/⁺ versus *Setd5*⁺/⁻; *P* = 0.052), effect of **α**-DTX on *Setd5*⁺/⁺ (saline versus **α**-DTX; *p* = 0.760) and *Setd5*⁺/⁻ (saline versus **α**-DTX; *p* = 0.649) and after **α**-DTX (*Setd5*⁺/⁺ versus *Setd5*⁺/⁻; *P* = 0.075). (**G**) Shelter exit behaviour during the first LER trial when the mice are injected with saline (*Setd5*⁺/⁺, black, top left; *Setd5*⁺/⁻, red, bottom left) or **α**-DTX (*Setd5*⁺/⁺, blue, top right; *Setd5*⁺/⁻, purple, bottom right). Each row represents 1 animal, filled and open dots represent exits crossed into the threat zone or not, respectively. (**H**) Effect of **α**-DTX on shelter exits. Before **α**-DTX (*Setd5*⁺/⁺ versus *Setd5*⁺/⁻; *P* = 0.012), effect of **α**-DTX on *Setd5*⁺/⁺ (saline versus **α**-DTX; *p* = 0.038) and *Setd5*⁺/⁻ (saline versus **α**-DTX; *p* = 0.003) and after **α**-DTX (*Setd5*⁺/⁺ versus *Setd5*⁺/⁻; *P* = 0.495). Markers represent the average values across all trials for individual animals. *P*-values: Wilcoxon's test, *p*-values: paired *t* test. The data underlying this figure can be found in S8 Data.

The dPAG hypoexcitability phenotype indicates that the integration and action initiation required for adequate perceptual decision-making is disrupted in *Setd5* animals. This physiological phenotype is tightly linked with behavioural impairments. Substantial threat evidence,

either by strong optogenetic activation of dmSC neurons or high-contrast visual looms, elicit a slower and less robust response in $Setd5^{+/-}$ compared to their WT siblings. On the other hand, limited threat evidence, either by weak optogenetic activation or low-contrast looms, instructs similar behavioural responses between genotypes (Fig 4). Accordingly, only strong current injections that match the expected dPAG drive caused by high-contrast loom [29] cause a Kv-channel induced hypoexcitability phenotype (Fig 5), indicating that the delayed response phenotype is due to a dPAG dysfunction. This stimulus strength relationship aligns with the coping difficulties of many individuals with ASD to salient sensory stimuli, such as bright lights or crowded places, but not to mellow sensory environments [6]. Recently, changes in dPAG excitability and the expression of defensive behaviours have also been identified in the $Nlgn3^{+/-}$ rat model of autism [42]. Notably, these rats exhibit the inverse behavioural and physiological phenotype to that observed in $Setd5^{+/-}$ mice, displaying stronger responses auditory fear conditioning, prolonged place avoidance and hyperexcitability in dPAG cells. These complementary results reflect the wide range of sensory sensitivities observed in the human population with ASD and reinforce the PAG as an important area to investigate disrupted sensory processing and its far-reaching effects in the core symptomatology associated with autism.

Given that the expression levels of Kv channels remain unaltered (Figs 5K and S8), the causes of the physiological hypoexcitability phenotype probably arise through homeostatic misregulation, either by altered cellular or subcellular channel localisation [33], posttranslational modifications [46], or interactions with auxiliary subunits [47]. Through these mechanisms, neuronal excitability could be modified by changing the effective number of Kv channels present in the membrane, or by modulating the gating or conductance of the channels, without changing the overall expression of the channels. Further experiments are required to determine the validity of these hypotheses. The link between potassium channels and autism is well established [48]. Genetic analyses of individuals with ASD uncovered deleterious mutations in potassium channels [48]. Notably, these mutations do not include Kv1.1. Nevertheless, autistic-like repetitive and social behaviours in the $Scn2a$ haploinsufficiency mouse model have been rescued in a Kv1.1-deficient background [49], supporting our results that enhancing and not disrupting Kv1.1's contribution can be a fundamental factor in ASD.

Sensory processing and perceptual abnormalities have been shown to co-occur with classical ASD diagnostic criteria and have been recently proposed as promising behavioural biomarkers of autism [4]. Here, we show that the innate LER provides a reliable and quantitative framework to systematically link behavioural traits with the underlying molecular changes. This is not a trivial task, as several possible changes could lead to delays in LER. The simplest explanation for these behavioural impairments is abnormalities in early visual processing. For example, if salient, high-contrast loom stimuli are relayed as low-contrast looms to dPAG neurons, these would be weakly activated, leading to delayed LERs [29]. Other mechanisms that could lead to weaker dPAG activation include changes in synaptic transmission, e.g., from dmSC to dPAG, imbalance in the excitatory–inhibitory ratio of the circuits involved [50], abnormal development of brain connectivity [51], or reduced excitability in dPAG neurons, among many others. Therefore, the exact molecular changes are likely to be model-dependent. We show that in all models tested, early visual processing (S4 and S5 Figs) and most intrinsic properties (S7 and S9 Figs) remain unaffected. The main difference observed is the dPAG hypoexcitability phenotype, which we were able to causally link to LER in the $Setd5$ model (Fig 5). In addition, we were able to correlate a similar hypoexcitability phenotype in the dPAG with LER in $Ptchd1$ animals, albeit through a different molecular mechanism. This already shows that different molecular perturbations lead to similar behavioural phenotypes, as further demonstrated by the weaker LER deficits in $Cul3$ animals that are independent of the intrinsic properties of the dPAG (S9 Fig). This level of dissection can lead to important insights that

may reveal therapeutic targets, as envisioned by precision medicine approaches [36]. Specifically, our study shows that in the *Setd5* haploinsufficient model, these behaviours are not necessarily developmental, as they can be pharmacologically ameliorated in adulthood.

In summary, our work links innate LER dysfunctions across diverse genetic mouse models of ASDs. Specifically, in the *Setd5* haploinsufficient model, we found that a key behavioural node, the PAG [52], functions as an interface between sensory, limbic, and motor circuits that, when disrupted, causally affects seemingly unrelated behaviours. This appears to be related to observations in people with ASD, where the severity of sensory processing impairments is related to the severity of core symptoms associated with autism [18–20]. Future studies designed to dissect the causal relationships between innate sensorimotor deficits, such as LER, and core ASD behavioural symptoms, such as social and communication difficulties, will be a revealing avenue to build a comprehensive view of ASD.

## Materials and methods

### Mice

The study was discussed and approved by the institutional ethics committee of the University of Veterinary Medicine Vienna and the Institute of Science and Technology Austria in accordance with good scientific practice guidelines and national legislation under license numbers BMWF-68.205/0023-II/3b/2014 and 66.018/0017-WF/V/3b/2017. Male and female adult $Setd5^{+/-}$, $Setd5^{+/fl}$::*Emx1*-Cre, $Setd5^{+/-}$::*VGluT2*-ires-Cre, $Cul3^{+/-}$, $Ptchd1^{Y/-}$ mice were housed under a 12-h light/dark cycle (lights on at 07:00), with food and water available ad libitum. The animals were housed in groups of 2 to 6 animals per cage and were tested during the light phase. Animals used for the in vivo electrophysiological recordings or freely moving optogenetics experiments were group housed before surgery and individually housed after the surgery. Mice were chosen based on genotypes. Sex-matched animal pairs of control–mutant siblings from the same litters were compared to decrease variance due to age, environment, and genetic background.

### Generation of Ptchd1$^{Y/-}$ mice

$Ptchd1^{Y/-}$ mice for analysis were generated using the CRISPR/Cas9 system, as described in [53]. In brief, in vitro transcribed sgRNAs targeting the exon 2 of *Ptchd1* were microinjected into zygotes of C57BL/6NRj mice together with Cas9 *mRNA*. Offspring were screened by PCR for deletions using primers spanning the target sites (5′-GTAGGGCTGGAATCATGAGG-3′, 5′-CACATCCTTTGGTGTGATGC-3′). Deletion of *Ptchd1* exon 2 was confirmed by Sanger sequencing analyses. The founder line was established from a female $Ptchd1^{+/-}$ mouse with a 1.26-kb deletion. Analysis of *Ptchd1* transcript level by RT-PCR confirmed impaired *Ptchd1* expression in brain, cerebellum, and kidney of the $Ptchd1^{Y/-}$ mice, compared to control mice.

### Behavioural procedures

**Experimental setup.**  All behavioural experiments were performed at mesopic light levels, in a square, black, IR-transparent Perspex box (W: 32 cm × L: 32 cm × H: 32 cm) with a shelter made of the same material (9 cm × 9 cm × 3 cm) positioned in one corner. One wall of the box was 2 cm shorter to allow for ventilation and for the optic fibre to pass through for optogenetic experiments. A plastic screen covered with a plastic sheet with a matte surface was placed as a lid on top of the box, and a modified Texas Instruments DLP projector (DLP CR4500EVM) back-projected a grey background. The blue LED was exchanged for a high-power UV-LED (ProLight 1W UV LED, peak 405 nm) to improve the differential stimulation

of S pigments. The experiments were recorded with a near-IR camera (acA1920-150um, Basler, 60 Hz) from below the centre of the arena. Video recording, visual and optogenetic stimulation were controlled using custom software written in MATLAB, python, and Arduino. The entire apparatus was housed within a sound-deadening, light-proof cabinet, which was illuminated by an infrared light-emitting diode (LED) lamp (SIL 616.21, Sanitas).

**Standard looming escape response (LER) protocol.** All animals were habituated to the test arena at least 1 day before testing and were allowed to explore the arena for at least 20 min. The initial 10 min of this exploration time was recorded and used to analyse the animals' exploratory behaviour (S1 Fig). For the looming avoidance response (LER: Figs 1, 2, 3, 6, S1, S2 and S3) paradigm, responses were tested over 5 consecutive days. On each test day, the animals were given 10 min to acclimatise to the arena, after which they were subjected to 3 test trials, with a 3-min interval between each trial. Each trial lasted 180 s, within which the central position of the mouse was tracked online (Python) and was used to trigger the visual stimuli in a closed-loop manner whenever the mouse crossed a designated threshold distance (10 cm) from the centre of the shelter, with a minimum stimulus interval of 30 s. A typical experiment lasted approximately 30 min and the entire arena was cleaned with 70% ethanol between animals.

**LER protocol with a food reward.** In preparation for the reward trials, after the fourth day of testing a small piece of banana chip was placed in the opposite corner of the arena to the shelter and the mice were free to explore the arena without triggering any loom stimuli. The animals were left to explore until they successfully acquired the banana chip, after which time both the mouse and the chip were placed back in their home cage. The reward trial then took place after the fifth day of testing. Another small piece of banana chip was again placed in the opposite corner of the arena and the animals were free to explore; however, this time, whenever the mouse passed the threshold distance from the shelter, a looming stimulus would start, this time with no interstimulus interval. For different contrast experiments, animals were presented with looming stimuli of 20%, 50%, or 98% contrast in a pseudorandom order upon crossing the trigger threshold. The background luminosity remained constant, while the luminance of the disc was increased to generate low contrast stimuli.

**LER protocol with different contrast stimuli.** Mice were similarly habituated to the arena as in the standard LER experiments; however, animals underwent a single session of 3 LER tests over 1 day. In contrast to the standard LER protocol, in these experiments, upon entering the threat zone, they were exposed to looming stimuli of 20%, 50%, or 98% contrast in a pseudorandom order. The background luminosity remained constant, while the luminance of the disc was increased to generate low contrast stimuli.

**In vivo optogenetic activation experiments.** The same behavioural arena was used for the optogenetic activation experiments except with the standard shelter exchanged for an open shelter consisting of a black piece of perspex (9 cm × 5 cm) positioned 10 cm above the floor. Mice were similarly acclimatised to the arena, as in the LER procedure, but with the fibre optic cable (200 μm, 0.48 NA, Doric Lenses) attached via a rotary joint (RJ1, ThorLabs) to allow for unrestrained movement and minimal handling of the animals. Since it has previously been shown [29] that increasing light intensity can be used as a proxy for the level of mSC, and hence dPAG, activation, we decided to systematically modulate the laser light intensity in different trials. Once the trial began, mice were photostimulated (473 nm, 10 light pulses of 10 ms at 10 Hz; Shanghai Dream Lasers Technology) if they passed the threshold distance (10 cm) from the shelter, with a minimum interstimulus interval of 30 s. The initial laser intensity was set to a low irradiance (0.01 to 0.1 mW mm$^{-2}$) that did not evoke an observable behavioural response, then increased to 0.5, 5, 10, 15, and 20 mW mm$^{-2}$ with at least 3 tests (180 s each, with a maximum of 5 trials per test) at each intensity. Two mice never elicited an observable

escape behaviour, and postprocessing histological analysis revealed mislocalised or the absence of viral expression in these animals, and they, along with their matched siblings, were excluded from the analysis.

**In vivo cannula experiments.** The same behavioural arena was used for the cannula experiments with α-dendrotoxin (α-DTX). Two weeks after mice were implanted with the cannula, they were acclimatised to the arena for 20 min and gently handled. Forty-eight hours after acclimatisation to the arena, the mice were attached to the infusion system (tubing (PlasticsOne C313CT), infusion cannula (C315IS-5/SPC, 33GA, 5 mm), guide cannula (PlasticsOne, C315GS-5/SF, 26GA, 5 mm), and a 0.5-µL manual syringe (Model 7000.5, #86250, Hamilton) and placed into an open cage to allow them to move freely during the infusion of 500 nL of saline with 0.1% bovine serum albumin (BSA) at 100 nl min$^{-1}$. The mice were left for a further 5 min before detaching them from the infusion system and placing them within the arena. The first 5 min the mice were within the arena was recorded to assess the baseline activity of these animals. A single trial of the standard LER trial was initiated 15 min after the start of infusion. The mice were reattached to the infusion system and α-DTX (500 nL, 500 nM in 0.1% BSA) was infused at 100nl min$^{-1}$ 48 h after the first LER trial with saline. The mice were again left for 5 min before being detached and placed into the arena. Baseline activity was assessed by recording the first 5 min within the arena, then 15 min after the start of the infusion, the standard LER trial was initiated. This time, there was a 3-min gap between LER trials and 3 trials were conducted, as in the standard LER test. After the last test, the mice were reattached to the infusion system and infused with 500 nL of a 10% solution of neurobiotin in sterile saline at 100 nl min$^{-1}$, left for 5 min, then detached and placed back into their home cages. The animals were killed 48 h after infusion with neurobiotin and later stained with Streptavidin-594 in order to confirm the location of the cannula implantation and infusion.

## Viruses

The viruses used in this study: for optogenetic activation, adeno-associated virus (AAV) AAV9-EF1a-DIO-hChR2(E123T)T159C)-EYFP-WPRE-hGH ($1 \times 10^{13}$ viral genomes per ml (vg ml$^{-1}$), Addgene 35509); for control experiments, AAV9-Syn-Flex-GCaMP6m-WPRE-SV40 ($1 \times 10^{13}$ vg/ml, Addgene 100838).

## Surgical procedures

Mice were anesthetised with an intraperitoneal (IP) injection of ketamine (95 mg kg$^{-1}$) and xylazine (4.5 mg kg$^{-1}$), followed by metamizol (20 mg kg$^{-1}$, IP), buprenorphine (0.05 mg kg$^{-1}$, IP), and meloxicam (2 mg kg$^{-1}$, subcutaneous (SC)). If needed, isoflurane (0.5% to2% in oxygen, 0.8 l min$^{-1}$) was used to maintain anesthesia. Fur on the top of the head was removed with an electric hair remover (Braun Precision Trimmer, PT 5010) and then the mice were placed in a stereotaxic frame (Model 962, Kopf Instruments) and fixed using the ear bars. Eyes were protected using Oleo Vital eye cream, and a topical analgesic (Xylocain 2% gel) was applied with a q-tip before cutting the skin. Craniotomies of approximately 1 mm diameter were made using 0.5 mm burrs (Fine Science Tools) and dental drill (Foredom, HP4-917), and viral vectors were delivered using pulled pipettes (World Precision Instruments, 1.4 mm OD, 0.53 mm ID, #504949) made with a micropipette puller (DMZ Zeitz-puller, Germany) and delivered using an automatic nanoinjector (World Precision Instruments, Nanoliter 2010) at 20 nl min$^{-1}$. The skin was closed after surgery using surgical glue (Vetbond, 3M, #1469SB).

**Optogenetics experiments.** Setd5$^{+/+}$::VGluT2-ires-Cre and Setd5$^{+/-}$::VGluT2-ires-Cre sex-matched sibling mice were injected with AAV9-DIO-ChR2-eYFP (see Viruses) into the left dmSC (75 to 100 nl, ML: −0.5, AP: −0.5 to −0.7, DV: −1.45 to −1.6, from bregma). Control

mice were injected with 100 nl of AAV9-Flex-GCaMP6m into the left dmSC at the same coordinates. One optic fibre (400 μm diameter, CFMC54L02, ThorLabs) was implanted 250 μm dorsal to the injection site. Fibres were affixed using light-curing glue (Optibond Universal, Kerr Dental) and dental cement (SuperBond C&B Kit, Hentschel-Dental).

**In vivo electrophysiological recordings.** Mice were implanted with a custom-designed head plate over an approximately 0.5 to 1 mm diameter craniotomy (left hemisphere, AP: 0.4 to −0.4, ML: 0.4 to 0.8 from lambda), and a second craniotomy (approximately 0.5 mm) was made anterior to bregma on the right hemisphere, where a reference gold pin was implanted and lowered until it touched the brain surface. Both were cemented to the skull using light-curing glue (Optibond Universal, Kerr Dental) and dental cement (SuperBond C&B Kit, Hentschel-Dental), with the headplate additionally fixed using Charisma Flow (Kulzer GmBH). The craniotomy was covered with Kwik-Cast (World Precision Instruments) to protect the underlying tissue before recording.

**In vivo cannula experiments.** Setd5$^{+/+}$ and Setd5$^{+/-}$ sex-matched sibling mice were implanted with a single infusion cannula (C315IS-5/SPC, 33GA, 5mm) above the dorsal PAG (ML: −0.5, AP: −0.8 to −1, DV: −1.5 to −1.65, from lambda). Fibres were affixed using light-curing glue (Optibond Universal, Kerr Dental) and dental cement (SuperBond C&B Kit, Hentschel-Dental), and the skin was closed around the implant with surgical glue (Vetbond, 3M, #1469SB).

## Visual stimuli

All stimuli were created and presented using custom-made scripts in MATLAB (MathWorks), using the Psychophysics Toolbox extensions [54], and frames were presented at a refresh rate of 60 Hz.

**Arena behavioural experiments.** The standard stimulus consisted of a dark disc on a grey background presented on the lid over the centre of the arena that reached a maximum size of 40 deg/visual angle. The disc expanded over 500 ms at a speed of 80 deg s$^{-1}$ and then remained at this size for 250 ms before disappearing and a new expanding disc appeared immediately. A maximum of 5 expanding discs were shown upon triggering the visual stimulus. A red rectangle was presented along one side of the projected image every fifth frame, onto an out-of-view section of the arena lid. An optical filter reflected this red light back towards a photodiode (PDA36A2, ThorLabs), and this information was then used for post hoc synchronisation of the behavioural data and the stimulus.

**Head-fixed in vivo electrophysiological experiments.** Sensory stimuli were made using the Psychophysics Toolbox extensions in MATLAB (MathWorks) and presented and synchronised using custom-made LabVIEW software. At each depth, the mice were exposed to one 20-min presentation of a full field shifting spatiotemporal white noise stimulus (see [55]) in order to assess receptive fields, and 2 repetitions of the remaining stimuli (5 × loom bouts, moving bars, moving gratings, full-field flash). Each presentation would start with a uniform grey screen (60 s, green and UV LED intensities were set to match the sun spectrum from a mouse opsin perspective), followed by the first presentation of the looming stimulus, then the other stimuli in a pseudorandom order separated by 30 s gaps of a grey screen. White noise "checker" stimuli were presented at 20 Hz update for 10 min. The checker size was 6.56 deg$^2$, and the entire grid was shifted between 0 and 3.25 deg in both x- and y-axes after every frame. Auditory monitoring of responses to a flashing point stimulus was used in real time to gauge the rough centre of the receptive fields of the neurons being recorded, and the location of the looming stimulus was positioned to overlap with this position. The looming stimulus consisted of 10 repetitions of 5 sequential, individual looming stimuli, the same stimulus previously

described and used in the behavioural experiments. These loom bouts would be presented with intervals of pseudorandom lengths between 10 and 30 s with a grey screen in between. To assess direction selectivity, the mice were presented with full-filed moving square gratings (1.5 cycles s$^{-1}$ temporal frequency and 0.3 cycles deg$^{-1}$ spatial frequency) or a wide dark bar (38 deg s$^{-1}$ speed, 6.5 deg width) in 4 orientations, drifting in 8 different directions (separated by 45 deg) repeated twice for each orientation. Full-field ON-OFF flashes were presented in blocks of 10 repetitions of 0.5 s OFF, 1 s ON, and 0.5 s OFF stimuli.

## In vivo electrophysiology

**Data acquisition.**   A 32-channel silicon probe (NeuroNexus, USA, A32-OM32) was used to record extracellular spikes from different depths of the SC in $Setd5^{+/+}$ ($n = 3$), $Setd5^{+/-}$ ($n = 3$), $Cul3^{+/+}$ ($n = 3$), $Cul3^{+/-}$ ($n = 3$), $Ptchd1^{Y/+}$ ($n = 3$), and $Ptchd1^{Y/-}$ ($n = 3$) sex-matched sibling mice. At least 24 h after surgery, mice were placed on a suspended spherical treadmill (20 cm diameter) and head-fixed within a toroidal screen covering 190˚ of visual angle, which was illuminated by a DLP-light projector (Blue LED exchanged for UV 405 nm, 60 Hz) reflecting off a spherical mirror in front of the screen (see diagram in S4A Fig). The behaviour of the animal was recorded by a near-IR camera (Basler, acA1920-150um, 30 Hz) from the reflection of the animal in an IR-reflective heat mirror positioned 30 cm from the face of the mouse and 35 cm from the camera. Prior to recording, the probe was coated with DiI (1 mM in ethanol, Merck, #42364) for later histological confirmation of the recording site, and a wire was connected to the ground pin for both an external reference and ground. The kwik-cast seal was removed, revealing the brain surface and the confluence of the overlying superior sagittal and the left transverse sinus. Care was taken to position the probe posteriorly and medially as possible without damaging the sinuses or other blood vessels. It was then zeroed at the brain surface and then slowly lowered into the overlying cortex and the SC at 1 mm s$^{-1}$ to depths between 1.2 and 2.2 mm. The recording was started 15 min after reaching the desired depth to allow for stabilisation. The probe was lowered to 3 to 4 depths per animal with 250 µm intervals, to sample from cells across the depth of the SC. Each recording took approximately 30 min, with the animal in the setup for approximately 180 min in total. Data were recorded at 20 kHz (PCIe-6321, National Instruments and Intan board RHD2000 Evaluation Board version 1.0).

## In vitro electrophysiology

Mice were deeply anesthetised via IP injection of ketamine (95 mg kg$^{-1}$) and xylazine (4.5 mg kg$^{-1}$), followed by transcardial perfusion with ice-cold, oxygenated (95% O$_2$, 5% CO$_2$) artificial cerebrospinal fluid (ACSF) containing (in mM): 118 NaCl, 2.5 KCl, 1.25 NaH$_2$PO$_4$, 1.5 MgSO$_4$, 1 CaCl$_2$, 10 Glucose, 3 Myo-inositol, 30 Sucrose, 30 NaHCO$_3$ (pH = 7.4). The brain was rapidly excised and coronal sections of 300 µm thickness containing the SC and PAG were cut using a Linear-Pro7 vibratome (Dosaka, Japan). Slices were left to recover for 20 min at 35˚C, followed by a slow cool down to room temperature (RT) over 40 to 60 min. After recovery, 1 slice was transferred to the recording chamber (RC-26GLP, Warner Instruments, Holliston, MA, USA) and superfused with ACSF containing 2 mM CaCl$_2$ and 20 µM bicuculline methiodide (Tocris) at a rate of 3 to 4 ml/min at RT. In experiments using 100 nM α-dendrotoxin (Alomone Labs, Jerusalem, Israel), 0.1% BSA was added to the ACSF. Glass pipettes (B150-86-10, Sutter Instrument, Novato, CA, USA) with resistances of 3 to 4 MΩ were crafted using a P1000 horizontal pipette puller (Sutter Instrument) and filled with internal solution containing (in mM): 130 K-Gluconate, 10 KCl, 5 MgCl$_2$, 5 MgATP, 0.2 NaGTP, 0.5 EGTA, 5 HEPES (pH 7.4) adjusted with KOH. Biocytin (0.2% to 0.3%) was added to the internal

solution for post hoc morphological reconstruction. Neurons of the dPAG or the dmSC were visually identified using an infrared differential interference contrast video system in a BX51 microscope (Olympus, Tokyo, Japan). Electrical signals were acquired at 20 to 50 kHz and filtered at 4 kHz using a Multiclamp 700B amplifier (Molecular Devices, San Jose, CA, USA) connected to a Digidata 1440A digitizer (Molecular Devices) with pClamp10 software (Molecular Devices). Spontaneous excitatory postsynaptic currents (sEPSCs) were recorded at −60 mV for at least 1 min. Intrinsic membrane properties and neuronal excitability were measured in current clamp mode by applying current steps from −20 to 200 pA for 1 s in 10 pA increments. To measure DTX-sensitive membrane currents, voltage steps from −80 to 50 mV in 10 mV increments were applied from a holding potential of −60 mV before and after a 10- to 15-min incubation in 100 nM α-dendrotoxin, and only 1 cell per slice was recorded. Access resistance was constantly monitored between protocols and recordings with access resistances exceeding 20 MΩ or with changes in access resistance or holding current by more than 20% were discarded. After recordings, the pipette was carefully withdrawn and the slice was transferred to 4% paraformaldehyde (PFA) in PBS solution.

## Proteomics

**Samples preparation.**   Six biological replicates from cortex, SC, and periaqueductal grey (PAG) tissue were dissected from *Setd5*$^{+/+}$ and *Setd5*$^{+/−}$ adult mice, immediately frozen in liquid nitrogen, and conserved at −70˚C until further processing. Tissues were Dounce homogenised 15 times in RIPA buffer (50 mM Tris-HCl (pH 8), 150 mM NaCl, 0.2% sodium dodecyl sulfate, 0.5% p/p sodium deoxycholate, 1% Triton X-100, 1× cOmplete EDTA-free protease inhibitor cocktail (Roche), and 1 mM phenylmethylsulfonyl fluoride (PMSF, Sigma) and centrifuged at 16,000$g$ 4˚C for 20 min. Supernatants were further collected and stored at −20˚C. Total protein quantity was measured with the bicinchoninic acid assay (BCA), and a total amount of 30 µg of protein per sample was used for mass spectrometry measurements. All samples were cleaned up by SP3 using a commercial kit (PreOmics GmbH, 50 mg of beads per sample) and then processed using the iST kit (PreOmics GmbH) according to the manufacturer's instructions. Tryptic digestion was stopped after overnight incubation, and cleaned-up samples were vacuum dried. Finally, samples were vacuum dried and then redissolved with 10 min sonication in the iST kit's LC LOAD buffer.

**LC-MS/MS analysis.**   Samples were analysed by LC-MS/MS on a nanoElute 2 nano-HPLC (Bruker Daltonics) coupled with a timsTOF HT (Bruker Daltonics). Samples were first concentrated over an Acclaim PepMap trap column (5.0 µm C18-coated particles, 0.5 cm * 300 µm ID, Thermo Fisher Scientific P/N 160454), then bound onto a PepSep XTREME column (1.5 µm C18-coated particles, 25 cm * 150 µm ID, Bruker P/N 1893476) heated at 50˚C and eluted over the following 125 min gradient: solvent A, MS-grade H$_2$O + 0.1% formic acid; solvent B, 100% acetonitrile + 0.1% formic acid; constant 0.60 nL/min flow; B percentage: 0 min, 2%; 100 min, 18%; 125 min, 35%, followed immediately by a 5-min plateau at 95%. MS method: M/Z range = 350 to 1,200 Th, ion mobility range = 0.7 to 1.4 1/K0; transfer time = 60 µs, pre-pulse storage time = 12 µs, ion polarity = positive, scan mode = dia-PASEF; TIMS parameters: ramp time = 100 ms, accumulation time = 100 ms; PASEF parameters: ms/ms scans = 10, total cycle time = 1.16721 s, charge range = 0 to 5, intensity threshold for scheduling = 2,500, scheduling target intensity = 20,000, exclusion release time = 0.4 min, reconsider precursor switch = on, current/previous intensity = 4, exclusion window mass width = 0.015 m/z, exclusion window v·s/cm$^2$ width = 0.015 V*s/cm$^2$.

Raw files were searched in DiaNN version 1.8.1 in library-free mode against a *Mus musculus* proteome obtained from UniProtKB. Match-Between-Runs was turned off. Fixed cysteine

modification was set to Carbamidomethylation. Variable modifications were set to Oxidation (M), Acetyl (protein N-term), and Phospho (STY). Data were filtered at 1% FDR. DiaNN's output was reprocessed using in-house R scripts, starting from the main report table. MS1 intensities were renormalized using the Levenberg–Marquardt procedure to minimise sample-to-sample differences. The long format main report table was consolidated into a wide format peptidoforms table, summing up quantitative values where necessary. Peptidoform intensity values were renormalized using the Levenberg–Marquardt procedure and corrected using the ComBat function from the sva package to remove the replicate-related batch effect. Protein groups were inferred from observed peptides and quantified using an in-house algorithm that (i) computes a mean protein-level profile across samples using individual, normalised peptido-form profiles ("relative quantitation" step); (ii) following the best-flyer hypothesis, normalises this profile to the mean intensity level of the most intense peptidoform ("unscaled absolute quantitation" step); for protein groups with at least 3 unique peptidoforms, only unique ones were used. Estimated expression values were log10-converted and renormalized using the Levenberg–Marquardt procedure. $P$-values were adjusted using the Benjamini–Hochberg (false discovery rate (FDR)) method. Average log10 expression values were tested for significance using a two-sided moderated $t$ test per samples group. Significance thresholds were calculated using the Benjamini–Hochberg procedure for FDR values of 1%, 5%, 10%, and 20%. For all tests, regulated protein groups were defined as those with a significant $P$-value and a log2 ratio greater than 5% of control-to-average-control ratios.

## Histology

For fluorescent immunostainings and histological confirmation of fibre and probe locations, adult $Setd5^{+/+}$ and $Setd5^{+/-}$ mice were transcardially perfused with 4% PFA and their brains were dissected, postfixed overnight in 4% PFA, washed with 1X PBS, dehydrated in a 30% sucrose solution, and sectioned with a freezing microtome (SM2010R, Leica) to 100 μm (for probe/fibre/cannula confirmation) and 40 μm (for immunostainings) thick slices. For histological confirmation of probe and fibre locations, the sections were rinsed in PBS, stained with 4′,6-diamidino-2-phenylindole (DAPI; 2 mM in PBS) and mounted on Superfrost glass slides in a mounting medium containing 1,4-diazabicyclooctane (DABCO; Sigma, D27802) and Mowiol 4–88 (Sigma, 81381). For Kv1.1 immunostainings, tissue was rinsed and then blocked for 1 h at RT in a buffer solution containing 10% normal donkey serum (Abcam, ab7475), 0.1% Triton X-100 in PBS. Primary antibodies (Kv1.1, Rabbit, 1:200; Rb-Af400, Frontier Institute) were diluted in a blocking buffer and incubated for 48 h at 4°C. The tissue was then rinsed 3 times in PBS and left overnight to wash at 4°C. Secondary antibodies (Alexa-594 Donkey anti-rabbit, 1:1,000; R37119, Thermo Fisher Scientific) were diluted in 1X PBS and incubated for 2 h at RT in the dark, before being washed 3 more times in 1X PBS before being counterstained with DAPI and mounted as for histological sections described above. Sections were imaged with either a confocal microscope (LSM800, Zeiss), at 63X or 20X magnification, or a wide-field microscope (VS120, Olympus), at 10X magnification.

For visualisation of the dPAG/dmSC cells recorded in vitro and the neurobiotin infused into the cannula implanted mice, tissue sections were washed in 1X PBS for 10 min, then a solution of Streptavidin (1:100, 1 mg/ml, Thermo Fisher Scientific, S11227) in PBS with 0.3% Triton-X was added and the samples were left at 4°C overnight. The tissue was then washed 3 times with PBS, and in vitro slices were mounted into a custom designed chamber for thick slices and imaged with a confocal microscope (LSM800, Zeiss) at 20X (90 μm projection, 2 μm sections) and 40X (50 μm projection, 1 μm sections).

## Western blot

Tissue samples from the SC, PAG, and cortex were dissected from $Setd5^{+/+}$ ($n$ = 3) and $Setd5^{+/-}$ ($n$ = 3) mice. PAG tissue was dissected using a 2-mm diameter biopsy punch (ID: 2 mm, OD: 3 mm, Fine Science Tools, 18035–02). Each purification was performed independently 3 times, and samples were stored at −80°C until further processing. Samples were Dounce homogenised 15 times with 300 mL RIPA buffer (50 mM Tris-HCl (pH 8), 150 mM NaCl, 0.2% sodium dodecyl sulphate, 0.5% p/p sodium deoxycholate, 1% Triton-X, 1x cOmplete EDTA-free protease inhibitor cocktail (Roche) and 1 mM PMSF (Sigma) using the loose pestle and sonicated for 10 s. The lysed samples were centrifuged for 20 min at 16,000$g$, and the supernatant was stored at −80°C. Total protein concentration was determined using the BCA protein assay (Pierce). Proteins were boiled for 5 min in Laemmli sample buffer before being separated at 200 V for 40 min using TGX stain-free precast gels (Bio-Rad). Stain-free protein gels were activated with UV light for 5 min and then transferred to PVDF membranes (Bio-Rad) using a turbo transfer system (Bio-Rad). Membranes were blocked for 1 h in TBST (Tris-buffered saline with 0.1% Tween 20) supplemented with 5% BSA (Sigma) and then incubated overnight at 4°C with anti-Kv1.1 primary antibody (Rb-Af400, Frontier Institute, 1:100). After primary antibody incubation, membranes were washed 4 times with TBST for 5 min and then incubated with anti-rabbit IRDye 800CW (LI-COR Biosciences, dilution 1:4,000) for 1 h at RT.

## Quantification and statistical analysis

**Analysis of behavioural data.**   For all behavioural experiments, the initial tracking and analysis of the escape behaviour were done blind through automatic analysis scripts, and the genotypes were added later when pooling the data across animals. Automatic image processing of the behavioural video frames identified the central position and the heading direction of the mouse.

**Standard looming escape response protocol.**   LERs were assessed by analysing a peristimulus time period starting from 3 s before the stimulus until 10 s after the stimulus onset. The speed of the animal was calculated as a moving mean over 83 ms time bins, and responses were classified as escapes if the mouse returned to the shelter within 7 s after stimulus onset and reached a maximum speed of at least 20 cm s$^{-1}$. Behavioural metrics were calculated by finding the mean value across all escape trials for each animal and then averaging across mice from each genotype. The maximum escape speed was calculated as the peak value of the speed trace between the onset of the escape and entry into the shelter. The reaction time of the mouse was calculated as the time from stimulus onset till the mouse reached the maximum speed during its escape back to the shelter. This reaction time was then used to group the responses of the animals depending on which loom presentation the peak of their escape response was recorded. The number of looms triggered is the total number of loom stimuli triggered across the 5 days of testing, not including the reward trial. To analyse the immediate change in speed following stimulus onset, trials were pooled across animals from each genotype. The difference between the average speed ±50 ms around the start of the stimulus ($S_{at}$) and the average speed in the time 300 to 800 ms after the start of the stimulus ($S_{im}$) was calculated separately for trials where the mouse initiated an escape within the presentation of the first looming stimulus or after the presentation of the first looming stimulus. Responses were categorised based on whether the $S_{im}$ was higher (>1 standard deviation (SD)), lower (<1 SD), or within 1 SD from the average speed at the time of triggering the stimulus ($S_{at}$) across all mice from the same genotype and the distribution of trials in each category was analysed using an $X^2$ test of independence.

**Exploratory behaviour and positional controls.**    Exploratory behavioural metrics during the acclimatisation period were calculated from videos of the first 10 min the mice were placed within the arena. A shelter exit was defined as an episode where the full body of the mouse moved from the inside to the outside of the shelter. The maximum distance travelled during an exit was defined as the maximal distance reached in a straight line between the centre of the mouse and the centre of the shelter. The exit duration was defined as the time between when the full body of the mouse crossed over until it recrossed the shelter boundary. The interbout interval during exploration was the time between subsequent shelter exits. The centre position of the mouse, [Xm, Ym], was used to quantify the percentage of the 10 min recorded that the mouse was either outside of the shelter, in the centre of the area (defined as a circular area, centred on the centre of the arena where the loom was presented, $[X_c, Y_c]$, with a 7-cm radius), or at the edge of the area (defined as outside of a circular area, centred on $[X_c, Y_c]$ with a 14-cm radius). To understand the effect of certain parameters of the mouse's position and location within the arena on its LER response, we calculated the distance of the mouse from shelter (defined as the straight line distance between $[X_m, Y_m]$ and $[X_s, Y_s]$) and the heading angle of the mouse at the time of stimulus start (defined as the angle (B C) between point B, the tip of the mouse's snout, A, the position between the 2 back paws of the mouse and C the centre of the shelter, $[X_s,Y_s]$). The position of the mouse's body parts was located using DeepLabCut tracking [56].

**Looming escape response protocol in the presence of a food reward.**    Food reward experiments were analysed with the same scripts as the standard LER experiments. When quantifying the reaction time and maximum speed, only trials where the animals met the escape criteria were included. Four $Setd5^{+/-}$ trials were removed from the final analysis because the mouse was interacting with the banana chip reward during the stimulus presentation. The number of looms and number of exits over the reward trials were calculated as the mean across animals from of the total per animal across genotypes.

**Different contrast experiments.**    Trials were grouped by stimulus contrast and pooled across animals from the same genotype. A repeated measure ANOVA analysis followed by a Bonferroni correction for multiple analyses was used to assess the effect of stimulus contrast on reaction time across genotypes, and individual $X^2$ tests of independence were used to assess the distribution of trials into the 3 response categories based on the change in immediate speed for each contrast. Only trials that fulfilled the escape criteria of having returned to the shelter within 7 s of stimulus onset and reaching a maximum speed of $>20$ cm s$^{-1}$ were included in the analysis.

**In vivo cannula experiments with α-DTX.**    Trials were pooled across animals from each genotype, and only trials that met the escape criteria used in the standard LER trials were included in the analysis of reaction time and maximum escape speed. Shelter exits were quantified as the average number of shelter exits per animal over the LER trials either with saline or with α-DTX. Comparisons of $Setd5^{+/+}$ or $Setd5^{+/-}$ animals with saline versus α-DTX were done with paired $t$ tests if the data reached the normality criteria; otherwise, paired Wilcoxon's rank sum tests were used. Unpaired Wilcoxon's rank sum tests were used for comparisons across genotypes.

## Analysis of optogenetics experiments

Trials were grouped according to the laser power used and pooled across animals from each genotype. As in the analysis of the LER experiments, the speed of animal was calculated across an 83-ms time bin, and the effect of laser stimulation was assessed by comparing the mean speed of the animal ±50 ms around the onset of the laser ($S_{at}$) and 300 to 800 ms after the onset

of the laser ($S_{im}$). Trials from the 0.01 mW mm$^{-2}$ condition were not subject to the escape criteria since laser stimulation at this intensity did not elicit an observed response; however, trials from all other laser power conditions had to meet the escape criteria used in the standard LER experiments in order to be included in further analyses. Trials were grouped by laser power intensity, and $S_{at}$ and $S_{im}$ were compared for all trials by paired Wilcoxon signed-rank tests. Similar to the LER analysis, responses were individually grouped based on both the sign and the magnitude of the difference between $S_{at}$ and $S_{im}$, into 3 groups where $S_{im}$ is either more than 1 SD greater than $S_{at}$, more than 1 SD less than $S_{at}$ or within 1 SD of $S_{at}$. Chi-squared tests were used to compare the distribution of the *Setd5$^{+/+}$* and *Setd5$^{+/-}$* trials at different laser powers.

For the control experiments, with GcaMP-injected mice (S6 Fig) the $Log_{10}(\Delta Speed)$ was calculated as the logarithmic of the ratio between $S_{im}$ and $S_{at}$. The change in $Log_{10}(\Delta Speed)$ across laser powers was analysed using a two-way repeated measures ANOVA looking at the effect of genotype, laser power and their interaction, followed by a multiple comparisons analysis with Bonferroni correction.

### Analysis of in vivo silicon probe recordings

**Preprocessing.**   Initial analysis was performed in MATLAB R2018b and later in MATLAB R2019a. Raw voltage traces were high-pass filtered (300 Hz), and spikes were detected and sorted using Kilosort2 (https://github.com/cortex-lab/Kilosort) and Phy as previously described [57], followed by manual curation. Firing rates histograms were calculated as the average firing rate across 16.7 ms time bins. Analog input channels (Photodiode, camera trigger) were extracted in MATLAB (MathWorks), and arduino were recorded at a 20-kHz sample rate and were used for offline synchronisation. Custom-made scripts in MATLAB were used to extract the time points of the visual stimulus onsets and matched them to the corresponding video camera frame.

**Determining the surface of the superficial superior colliculus (sSC).**   Initial SC location was determined in real time by auditory monitoring of multiunit activity to a flashing stimulus and was later confirmed offline using a current source density analysis of the raw recording (see [31,58]) combined with histological reconstruction of the probe location visualised with DiI. CSD profiles were calculated for each recording and were used to align the depth of the units across recordings and across animals. The transition point between an overlying source and underlying sink was taken as the boundary between the stratum griseum superficiale (SGS) and the stratum opticum (SO). The boundary between the superficial (sSC) and intermediate/deep superior colliculus (idSC) was defined as 100 μm below this inflection depth (ID), taking into account the thickness of the SO. sSC was defined as 300 μm above the ID to 100 μm below the ID, and idSC was defined as the area below 100 μm below the ID. Only units identified as within the sSC and units with at least 2 spikes within the stimulus presentation were used for analysis. For all analyses, spikes were sorted into peristimulus periods based on synchronisation with the stimulus frames, and firing rate histograms were calculated over 16.7 ms time bins.

**Analysis of baseline firing properties.**   Responses to a full field grey screen during the first 60 s of recording were taken to determine the baseline firing properties of the recorded cells. The baseline firing rate per animal was calculated as the mean firing rate across cells in the 1,000 μm below the sSC surface.

**Analysis of responses to flash stimulus.**   Responses to a single flash stimulus (0.5 s OFF; 1 s ON; 0.5 s OFF) were generated by averaging the firing rate histograms across 10 repetitions of the stimulus. The responsiveness of single units was defined based on the *p*-value generated

from running a ZETA test [59] on the average firing rate histograms using the 30-s period before the stimulus as baseline and the 2-s flash stimulus as the response time. Clustering of responsive units was done using K-means, and the number of clusters was determined by identifying the elbow point when analysing the Calinski–Harabasz index. Clustering was done with units from both genotypes pooled together and then later split based on genotype. The proportion of flash-responsive cells for *Ptchd1* and *Cul3* animals were determined for each animal by dividing the number of sSC units that were $p < 0.05$ by the total number of sSC units found.

**Analysis of responses to loom stimulus.** Responses to a single loom stimulus were obtained by averaging the responses to the first loom of the loom bout stimulus (consisting of 5 consecutive looms) across 10 repetitions. Responsiveness was again defined as units having a *p*-value (determined using the ZETA test) of $< 0.05$, with the 30 s preceding the loom stimulus as baseline and the 0.7 s of first loom presentation as the response time. The average firing rate was calculated as the overall average of the mean firing rate recorded across each of the 5 loom presentations individually for 10 repetitions. Maximum firing was similarly calculated as the mean maximum firing during each of the 5 looms, averaged across 10 repetitions. The time-to-peak was calculated as the mean of the time to the peak firing within each of the 5 looms, averaged over the 10 repetitions.

**Analysis of responses to shifting white noise stimulus.** To further analyse the visual receptive fields of the units recorded, we analysed their responses to a shifting white noise stimulus [55] and calculated them as spike-triggered averages. The location of the centre of the receptive field was estimated by finding the pixel with the highest variance over a peri-spike time period, and the final receptive field was cropped to $160 \times 160$ pixels centred on this pixel with highest variance. Signal-to-noise ratio (SNR) for each neuron is defined as $10 \cdot log_{10}(max_{power}/noise_{power})$, where $max_{power}$ is the square of the pixel value at the receptive field centre, and $noise_{power}$ the variance of all the values in a 5 pixels border around the edge of the receptive field. Visually responsive neurons were defined as those with SNRs in the 80th percentile of the SNRs of all sSC units or above. The centre of the receptive field was assigned as ON or OFF based on the sign of the mean of the central pixel at the centre of the receptive field. The proportion of OFF receptive fields was calculated as the ratio between visually responsive cells with OFF receptive fields and the total number of visually responsive cells in the sSC. The radius of the receptive field centre was calculated as the width at half-peak of the spatial receptive field profile. The average radius of the receptive fields of an animal was calculated using only the visually responsive units. Temporal receptive fields were clustered similarly to flash responses, using K-means to cluster the responses, and the number of clusters was determined using the Calinski–Harabasz index.

**Analysis of responses to moving bars and gratings stimuli.** For each direction, the mean across trials was calculated and normalised to the baseline firing in the 30 s before stimulus onset. The preferred direction ($D_p$) was defined as the direction with the greatest normalised response, and the nonpreferred direction ($D_{np}$) was taken as 180 degrees from that angle. The direction selectivity index (DSI) was then computed as ($D_p - D_{np}$) / ($D_p + D_{np}$) with values falling in a range from 1 (high direction selectivity) to 0 (low direction selectivity). Units were defined as being "direction selective" (DS) if their DSI was in the 80th percentile or higher out of all units recorded within the sSC. The proportion of DS cells per animal was found by quantifying the proportion of DS sSC units divided by all sSC units recorded for that animal.

**Analysis of behavioural data from head-fixed visual stimulation.** The motion of the animal was determined by sorting the difference between pixels in neighbouring frames with K-means into 2 groups. The proportion of time each animal spent running was calculated as the proportion of frames that were registered as "moving" over the total number of frames.

The pupillary light reflex (PLR) of the animals was assessed by tracking the change in pupil size across the repetitions of the flash stimulus. A deep neural network approach for markerless tracking [56] was used to extract points around the pupil. The diameter of the pupil was then calculated by fitting an ellipse to these points. Since we observed, and it is known, that the size of the pupil is greatly affected by locomotion, when analysing the PLR, we only included repetitions in which the mice were running. An animal was considered to be running during a repetition if the mouse was classified as "moving" in over half of the video frames covering that repetition.

## Analysis of in vitro patch clamp electrophysiological recordings

Whole-cell recordings were analysed in Clampfit (Molecular Devices) and MiniAnalysis Program (Synaptosoft). Resting membrane potential (RMP) was measured as the average membrane potential prior to current injection. Input resistance ($R_{in}$) was calculated from the slope of the linear fit of the hyperpolarising current/voltage relationship. Membrane time constant $\tau$ was measured from the exponential fit of the membrane potential change in response to a –10 or –20 pA current injection. Membrane capacitance ($C_m$) was calculated from $C_m = \tau/R_{in}$. Action potential firing was quantified in Clampfit using the event detection function. Phase-plane plots were performed to investigate the dynamic changes in the membrane potential over time (dV/dt) and were generated using the spikes from the first current step necessary to elicit action potentials (rheobase current). To visualize DTX-sensitive currents, 2 to 3 voltage-step protocols under baseline and DTX conditions were averaged and then the average current trace after DTX incubation was subtracted from the average baseline current trace. Currents were quantified as the maximal current deflection during the 1-s voltage step in either direction. To correct for different neuronal membrane surface areas between cells, current responses were divided by Cm and expressed as current density (pA/pF). Soma size was calculated by fitting an oval to the projected image (63X) of the neurons and calculating the area of that oval.

## Analysis of proteomics

MS files were searched in DiaNN version 1.8.1 in library-free mode against a *Mus musculus* proteome obtained from UniProtKB. Match-Between-Runs was turned off. Fixed cysteine modification was set to Carbamidomethylation. Variable modifications were set to Oxidation (M), Acetyl (protein N-term) and Phospho (STY). Data were filtered at 1% FDR. DiaNN's output was reprocessed using in-house R scripts, starting from the main report table. MS1 intensities were renormalized using the Levenberg–Marquardt procedure to minimize sample-to-sample differences. The long format main report table was consolidated into a wide format peptidoforms table, summing up quantitative values where necessary. Peptidoform intensity values were renormalized using the Levenberg–Marquardt procedure and corrected using the ComBat function from the sva package to remove the replicate-related batch effect. Protein groups were inferred from observed peptides and quantified using an in-house algorithm that (i) computes a mean protein-level profile across samples using individual, normalized peptidoform profiles ("relative quantitation" step); (ii) following the best-flyer hypothesis, normalizes this profile to the mean intensity level of the most intense peptidoform ("unscaled absolute quantitation" step); for protein groups with at least 3 unique peptidoforms, only unique ones were used; otherwise, razor peptidoforms were also included; phosphopeptidoforms and their unmodified counterparts were excluded from the calculations. Estimated expression values were log10-converted and renormalised using the Levenberg–Marquardt procedure. *P*-values were adjusted using the Benjamini–Hochberg (FDR) method. Average log10 expression values

were tested for significance using a two-sided moderated *t* test per samples group and a global F-test (limma). Significance thresholds were calculated using the Benjamini–Hochberg procedure for FDR values of 1%, 5%, 10%, and 20%. For all tests, regulated protein groups were defined as those with a significant *P*-value and a log2 ratio greater than 5% of control-to-average-control ratios. GO terms enrichment analysis was performed comparing for each test regulated against observed protein groups (topGO). Statistical analysis was performed for phospho-modified peptides as for protein groups, normalizing ratios to parent protein group (s).

## Analysis of histological data

For comparison between the Kv1.1 expression in *Setd5*$^{+/+}$ and *Setd5*$^{+/−}$ tissue 30 μm stacks were imaged with 1 μm sections at 63X magnification, or 55 μm stacks were imaged with 2 μm sections at 20X magnification, and the same laser power settings were used for all images at a given magnification.

## Analysis of western blot data

The Bio-Rad ChemiDoc MP system was used for imaging and acquisition and images were quantified using ImageJ software. Protein levels are displayed as fold change. Student *t* test was used to test for significance (MATLAB, MathWorks). Data represent mean ± SEM obtained from 3 mice per genotype.

## Statistics

Analyses were performed in custom-written MATLAB (2019a) scripts (MathWorks). Shapiro–Wilk and Levene's tests were used to test the normality and the equal variances of the datasets, respectively, and parametric or nonparametric tests were used accordingly. Two-tailed tests were used unless specified. All statistical tests are reported in the text and appropriate figure legends (*$p < 0.05$, **$p < 0.01$, ***$p < 0.001$). In bar plots, the mean ± SEM are shown, unless otherwise stated. Boxplots display the median and IQR.

## Supporting information

**S1 Fig. Additional *Setd5* behavioural controls.** No significant difference in the exploratory behaviour of the *Setd5*$^{+/+}$ and *Setd5*$^{+/−}$ animals during the prestimulus exploration. The top graphic depicts the dimensions and regions of the arena that are classified as the centre (pink, a circular region with a radius of 7 cm from the centre of the arena [$X_c$,$Y_c$]), and the edge (blue, the area outside of a circular area with a 14-cm radius from [$X_c$, $Y_c$]). (**A**) Average speed during prestimulus exploration (*Setd5*$^{+/+}$, 7.50 cm s$^{-1}$; *Setd5*$^{+/−}$, 6.65 cm s$^{-1}$, $P = 0.151$). (**B**) Maximum speed (*Setd5*$^{+/+}$, 42.2 cm s$^{-1}$; *Setd5*$^{+/−}$, 40.2 cm s$^{-1}$, $P = 0.642$). (**C**) Time spent out of the shelter (*Setd5*$^{+/+}$, 53.5%; *Setd5*$^{+/−}$, 40.9%, $P = 0.062$). (**D**) Time spent in the centre of the arena (*Setd5*$^{+/+}$, 6.00%; *Setd5*$^{+/−}$, 4.20%, $P = 0.448$). (**E**) Time spent at the edge of the arena (*Setd5*$^{+/+}$, 13.9%; *Setd5*$^{+/−}$, 9.47%, $P = 0.076$). (**F**) Number of shelter exits (*Setd5*$^{+/+}$, 8.14; *Setd5*$^{+/−}$, 7.88, $P > 0.999$). (**G**) Mean distance travelled during exit (*Setd5*$^{+/+}$, 15.2 cm; *Setd5*$^{+/−}$, 15.1 cm, $P = 0.981$). The lower graphic depicts the distance of the mouse from the shelter as the Euclidean distance between the centre of mouse, [$X_m$, $Y_m$], and the centre of shelter, [$X_s$, $Y_s$], and the heading angle as the angle (a˚) between a line drawn along the body axis of the mouse and another from the position between the back paws of the mouse and the centre of the shelter. The bottom graphic depicts how the directedness of the escape trajectory is calculated by finding the shortest Euclidean distance between the centre of the mouse and the edge

of the shelter when the stimulus starts and the actual distance travelled between the stimulus start and the mouse reentering the shelter. (**Hi**) Median distance from the shelter when the looming stimulus is triggered (*Setd5*$^{+/+}$, 13.2 cm; *Setd5*$^{+/-}$, 13.7 cm, $P = 0.341$). (**Hii**) Distribution of distances from the shelter across all trials, pooled by genotype ($p = 0.325$, two-way Kolmogorov–Smirnov test). (**Hiii**) Relationship between the distance to the shelter and the reaction time (*Setd5*$^{+/+}$, $p = 0.443$; *Setd5*$^{+/-}$, $p = 0.778$). (**Hiv**) Relationship between the distance to the shelter and the maximum escape speed reached during escape trials (black, *Setd5*$^{+/+}$, $r = 0.51$, $p < 0.001$; red, *Setd5*$^{+/-}$, $r = 0.16$, $p = 0.016$). (**Ii**) Median instantaneous speed when the looming stimulus is triggered (*Setd5*$^{+/+}$, 15.1 cm s$^{-1}$; *Setd5*$^{+/-}$, 12.0 cm s$^{-1}$, $P = 0.312$). (**Iii**) Distribution of the instantaneous speeds across all trials, pooled by genotype ($p = 0.086$, two-way Kolmogorov–Smirnov test). (**Iiii**) Relationship between the instantaneous speed and the reaction time (*Setd5*$^{+/+}$, $p = 0.197$; *Setd5*$^{+/-}$, $p = 0.112$). (**Iiv**) Relationship between the instantaneous speed and the maximum escape speed reached during escape trials (*Setd5*$^{+/+}$, $p = 0.592$; *Setd5*$^{+/-}$, $p = 0.580$). (**Ji**) Median heading angle when the looming stimulus is triggered (*Setd5*$^{+/+}$, 114°; *Setd5*$^{+/-}$, 111°, $P = 0.194$). (**Jii**) Distribution of heading angles across all trials, pooled by genotype ($p = 0.994$, two-way Kolmogorov–Smirnov test). (**Jiii**) Relationship between the instantaneous speed and the reaction time (*Setd5*$^{+/+}$, $r = -0.37$, $p = 0.014$; *Setd5*$^{+/-}$, $p = 0.333$). (**Jiv**) Relationship between the instantaneous speed and the maximum escape speed reached during escape trials (*Setd5*$^{+/+}$, $p = 0.635$; *Setd5*$^{+/-}$, $p = 0.188$). (**K**) Significant increase in speed upon stimulus presentation when mice respond within the first loom presentation (*Setd5*$^{+/+}$, 57 trials, $p < 0.001$; *Setd5*$^{+/-}$, 38 trials, $p < 0.001$, paired $t$ test). S$_{at}$ is the mean speed of the animal ±50 ms of stimulus onset, and S$_{im}$ is the mean speed of the animal 300-800ms after stimulus onset. Significant decrease in speed for *Setd5*$^{+/-}$ trials where the mice respond after the first loom (within second loom: 35 trials; within third loom: 31 trials; within fourth loom: 31 trials; within fifth loom: 16 trials; after fifth loom: 45 trials, all $p < 0.001$, paired $t$ tests). (**L**) Left, raster plot of mouse speed during escape trials for male *Setd5*$^{+/+}$ (top, $n = 4$, 18 trials) and *Setd5*$^{+/-}$ (bottom, $n = 4$, 144 trials) mice, sorted by reaction time. White, dotted vertical lines denote the start of each loom; white solid line denotes the end of the stimulus. Right, summary of the proportion of trials in which the mice respond *within each loom* for *Setd5*$^{+/+}$ (top) and *Setd5*$^{+/-}$ mice (bottom). Proportion of trials where mice escaped within the first loom: Setd5$^{+/+}$, 1.00, Setd5$^{+/-}$, 0.324, $P = 0.028$. Distribution of number of looms to escape, *Setd5*$^{+/+}$, 18 trials, *Setd5*$^{+/-}$, 144 trials, $p < 0.001$. (**M**) Summary of the average reaction time per animal (Setd5$^{+/+}$, 18 trials, 0.394 s; *Setd5*$^{+/-}$, 130 trials, 2.10 s, $P = 0.029$). (**N**) Summary of the average maximum escape speed per animal (Setd5$^{+/+}$, 18 trials, 63.3 cm s$^{-1}$; *Setd5*$^{+/-}$, 130 trials, 49.9 cm s$^{-1}$, $P = 0.047$). Box-and-whisker plots show median, IQR, and range. Error bars represent the SEM. *P*-values: Wilcoxon's test, *p*-values: Pearson's correlation, unless specified. Plotted linear fits depict the statistically significant correlations. The data underlying this figure can be found in S9 Data.
(TIF)

**S2 Fig. WT responses in cortical and hippocampal Setd5$^{+/-}$ animals.** (**A**) Raster plot of mouse speed in response to the looming stimuli (white, dotted vertical lines denote the start of each loom; white solid line denotes the end of the stimulus) for *Setd5*$^{+/+}$; *Emx1*-Cre (top, $n = 6$, 42 trials) and *Setd5*$^{+/fl}$; *Emx1*-Cre (bottom, $n = 6$, 32 trials), sorted by reaction time. Bottom, distribution of number of looms to escape across all trials (*Setd5*$^{+/+}$; *Emx1*-Cre, 32 trials, *Setd5*$^{+/fl}$;*Emx1*-Cre, 42 trials, $p = 0.960$, two-sample Kolmogorov–Smirnov (KS) test). (**B**) Pictorial representation of the expression of *Emx1* in a sagittal section of a mouse brain, showing the localisation of the expression to the cortex and hippocampus (blue, top). Proportion of trials in which the mice respond within each loom for *Setd5*$^{+/+}$; *Emx1*-Cre (middle) and *Setd5*$^{+/fl}$;

*Emx1*-Cre mice (bottom). (**C**) Reaction times and (**D**) maximum escape speed (*Setd5*$^{+/+}$; *Emx1*-Cre, $n = 6$, 61.1 cm s$^{-1}$; *Setd5*$^{+/fl}$;*Emx1*-Cre, $n = 6$, 66.4 cm s$^{-1}$, $P = 0.295$). (**E**) Speed immediately following the stimulus presentation for trials where the mice escape within the first loom presentation (left, *Setd5*$^{+/+}$; *Emx1*-Cre, $n = 6$, $p < 0.001$; right, *Setd5*$^{+/fl}$;*Emx1*-Cre, $n = 6$, $p < 0.001$). (**F**) Speed immediately following the stimulus presentation for trials where the mice escape after the first loom presentation (left, *Setd5*$^{+/+}$, $n = 3$, $p = 0.6248$, paired *t* test; right, *Setd5*$^{+/-}$, $n = 2$, $p = 0.7546$). *P*-values: Wilcoxon's ranked-sum test. *p*-values: two-tailed paired *t* test, unless specified. The data underlying this figure can be found in S10 Data. (TIF)

**S3 Fig. Ptchd1 animal model generation and Ptchd1 and Cul3 behavioural characterisation.** (**A**) CRISPR-Cas9 design for the generation of *Ptchd1*$^{Y/-}$ mice. (**B**) gDNA confirmation of the excised DNA (2.1 Kb in *Ptchd1*$^{Y/+}$ and 0.7 Kb in *Ptchd1*$^{Y/-}$). (**C**) Confirmation of loss of Ptchd1 cDNA in different tissues in *Ptchd1*$^{Y/-}$ mice, and control *Gapdh* presence throughout. (**D**) Exploration controls for *Ptchd1*. (**Di**) Average speed during prestimulus exposure acclimatisation (Ptchd1$^{Y/+}$, 7.53 cm s$^{-1}$; Ptchd1$^{Y/-}$, 9.31 cm s$^{-1}$, $p = 0.003$). (**Dii**) Time spent in the centre of the arena (Ptchd1$^{Y/+}$, 4.46%; Ptchd1$^{Y/-}$, 5.86%, $p = 0.170$). (**Diii**) Time spent at the edge of the arena (Ptchd1$^{Y/+}$, 14.3%; Ptchd1$^{Y/-}$, 13.0%, $p = 0.148$). (**Div**) Number of shelter exits (*Ptchd1*$^{Y/+}$, 17.0, *Ptchd1*$^{Y/-}$, 29.5, $p = 0.003$). (**Dv**) Average distance travelled during exit (*Ptchd1*$^{Y/+}$, 17.1 cm, *Ptchd1*$^{Y/-}$, 14.6 cm, $p = 0.004$). (**E**) Speed change immediately following the stimulus presentation for trials where the mice escape within the first loom presentation (top left, *Ptchd1*$^{Y/+}$, $n = 9$, black, $p < 0.001$, two-tailed *t* test; top right, *Ptchd1*$^{Y/-}$, $n = 9$, red, $p < 0.001$, two-tailed *t* test). Speed change immediately following the stimulus presentation for trials where the mice escape after the first loom presentation (bottom left, *Ptchd1*$^{Y/+}$, $n = 1$, black; bottom right, *Ptchd1*$^{Y/-}$, $n = 8$, red, $p < 0.001$, two-tailed *t* test). S$_{at}$ is the mean speed of the animal ±50 ms of stimulus onset, and S$_{im}$ is the mean speed of the animal 300-800ms after stimulus onset. (**F**) Top, relationship between reaction time and test day (*Ptchd1*$^{Y/+}$, $r = 0.029$, $p = 0.840$; *Ptchd1*$^{Y/-}$, $r = 0.317$, $p < 0.001$) and bottom, maximum escape speed and test day (*Ptchd1*$^{Y/+}$, $r = -0.254$, $p = 0.07$; *Ptchd1*$^{Y/-}$, $r = -0.040$, $p = 0.556$). (**G**) Mean reaction time (left) and maximum escape speed (right) per animal, for the very first loom presentation (reaction time; *Ptchd1*$^{Y/+}$, 0.423 s, *Ptchd1*$^{Y/-}$, 1.39 s, $P = 0.014$; maximum escape speed; *Ptchd1*$^{Y/+}$, 69.1 cm s$^{-1}$, *Ptchd1*$^{Y/-}$, 57.3 cm s$^{-1}$, $P = 0.139$). (**H-J**) Relationship between test day and the number of exits from shelter (**H**, *Ptchd1*$^{Y/+}$, $r = 0.122$, $p = 0.134$; *Ptchd1*$^{Y/+}$, $r = 0.101$, $p = 0.1936$), the maximum distance travelled (**I**, *Ptchd1*$^{Y/+}$, $r = -0.213$, $p = 0.071$; *Ptchd1*$^{Y/+}$, 112 trials, $r = 0.103$, $p = 0.278$) and the duration of time spent outside of the shelter (**J**, *Ptchd1*$^{Y/+}$, $r = 0.148$, $p = 0.214$; *Ptchd1*$^{Y/+}$, $r = 0.286$, $p = 0.002$). (**K**) Exploration controls for *Cul3*. (**Ki**) Average speed during prestimulus exposure acclimatisation. Cul3$^{+/+}$, 7.60 cm s$^{-1}$; Cul3$^{+/-}$, 8.23 cm s$^{-1}$, $p = 0.3055$. (**Kii**) Time spent in the centre of the arena (Cul3$^{+/+}$, 7.10%; Cul3$^{+/-}$, 8.32%, $p = 0.532$). (**Kiii**) Time spent at the edge of the arena (Cul3$^{+/+}$, 12.2%; Cul3$^{+/-}$, 13.6%, $p = 0.480$). (**Kiv**) Number of shelter exits (Cul3$^{+/+}$, 13.3, Cul3$^{+/-}$, 14.7, $p = 0.526$). (**Kv**) Average distance travelled during exit (Cul3$^{+/+}$, 15.8 cm; Cul3$^{+/-}$, 15.7 cm, $p = 0.945$). (**L**) Top, speed change immediately following the stimulus presentation for trials where the mice escape within the first loom presentation (top left, *Cul3*$^{+/+}$, $n = 10$, black, $p < 0.001$, two-tailed *t* test; top right, *Cul3*$^{+/-}$, $n = 9$, pink, $p = 0.008$, two-tailed *t* test). Speed change immediately following the stimulus presentation for trials where the mice escape after the first loom presentation (bottom left, *Cul3*$^{+/+}$, $n = 5$, black, $p = 0.017$; bottom right, *Cul3*$^{+/-}$, $n = 8$, pink, $p = 0.001$, two-tailed *t* test). (**M**) Top, relationship between reaction time and test day (*Cul3*$^{+/+}$, $r = 0.3146$, $p = 0.012$; *Cul3*$^{+/-}$, $r = 0.109$, $p = 0.233$); bottom, relationship between maximum escape speed and test day (*Cul3*$^{+/+}$, $r = -0.2679$, $p = 0.033$; *Cul3*$^{+/-}$, $r = 0.125$, $p = 0.1711$). (**N**) Mean reaction

time (left) and maximum escape speed (right) per animal, for the very first loom presentation (reaction time; $Cul3^{+/+}$, 0.446 s, $Cul3^{+/-}$, 1.26 s, $P = 0.117$; maximum escape speed; $Cul3^{+/+}$, 67.5 cm s$^{-1}$, $Cul3^{+/-}$, 45.8 cm s$^{-1}$, $P = 0.003$). (**O, P**) Relationship between test day and the number of exits from shelter (**O**, $Cul3^{+/+}$, $r = 0.083$, $p = 0.289$; $Cul3^{+/-}$, $r = -0.1198$, $p = 0.1253$), the maximum distance travelled (**P**, $Cul3^{+/+}$, $r = -0.081$, $p = 0.480$; $Cul3^{+/-}$, $r = -0.224$, $p = 0.034$) and the duration of time spent outside of the shelter (**Q**, $Cul3^{+/+}$, $r = -0.036$, $p = 0.754$; $Cul3^{+/-}$, $r = -0.091$, $p = 0.393$). Box-and-whisker plots show median, IQR, and range. $p$-values: Wilcoxon's test, $P$-values: Pearson's correlation analysis, unless specified. Trend lines are only drawn for significant correlations. The data underlying this figure can be found in S11 Data and S12 Data.
(TIF)

**S4 Fig. Intact visual processing across the superior colliculus.** (**A**) Schematic of the in vivo recording setup displaying the spherical treadmill and light projector illuminating the spherical screen. (**B**) Left, schematic of the 32-channel silicon probe used to record extracellular activity aligned with the current source density analysis of a single flash stimulus averaged over 10 presentations. Black vertical line represents the stimulus onset; yellow dotted line marks the inflection depth separating the current source and sink. (**C**) Histological reconstruction of the probe position marked with DiI (120 μm coronal section, scale bar: 1 mm) with a close-up (scale bar: 250 μm). Yellow dotted line as in (**B**). (**D**) Proportion of time the mice were moving during the recording sessions ($P = 0.400$, top) and SC's baseline firing rate ($P = 0.880$, bottom). (**E**) Pupil dilation in response to the full-field flash stimulus. Example images of dilated and constricted pupils in responses to the OFF (left) and ON (right) periods of the flash. $Setd5^{+/+}$ (black), $Setd5^{+/-}$ (red) ($P = 0.060$). (**F**) Sorted raster plot of neural responses to a single flash stimulus. Vertical red dotted lines indicate the on and offset of the flash stimulus. (**G**) Proportion of cells in each cluster (top). Mean ± SEM of the flash responses in each cluster (traces). (**H**) Proportion of direction-selective SC cells (see **Materials and methods**) to full field gratings ($Setd5^{+/+}$, 87 of 153 units, 0.287; $Setd5^{+/-}$, 159 of 259 units, 0.2432, $P = 0.322$, Wilcoxon's test). (**I_i**) Firing rate of example units in response to full field gratings moving in 8 different directions for $Setd5^{+/+}$ (black) and $Setd5^{+/-}$ (red) with a summary polar plot of their direction selectivity (right) and the corresponding direction selectivity index (DSI) value. Bold lines show the mean response across 3 repetitions (light grey lines). (**I_ii**) Distribution of DSI across SC units (top, $Setd5^{+/+}$, 153 units, median DSI = 0.323; bottom, $Setd5^{+/-}$, 259 units, median DSI = 0.3018, $P = 0.7054$, two-sample Kolmogorov–Smirnov test). (**J**) Mean and spike raster plots from single $Setd5^{+/+}$ (black) and $Setd5^{+/-}$ (red) units to 10 repetitions of a single loom stimulus. (**K-M**) Summary of mean, maximum, and time-to-peak firing (**K**, $P = 0.215$; **L**, $P = 0.100$; **M**, $P = 0.372$). (**N**) Mean ± SEM response of all and (**O**) of the 5 consecutive loom stimuli for $Setd5^{+/+}$ (black) and $Setd5^{+/-}$ (red) units. (**P**) Average firing to a single loom across the 5-loom stimulus ($p = 0.732$). Box-and-whisker plots show median, IQR, and range. $P$-values: Wilcoxon's test, $p$-values: two-way repeated measures ANOVA, unless specified. The data underlying this figure can be found in S13 Data.
(TIF)

**S5 Fig. Visual response properties for Cul3 and Ptchd1.** (**A**) Schematic of visual stimulus and proportion of sSC cells that are flash responsive ($Ptchd1^{Y/+}$: 0.350; $Ptchd1^{Y/-}$: 0.343, $P = 0.857$, two-way $t$ test). (**B**) Sorted raster plot of neural responses to a single flash stimulus. Vertical red dotted lines indicate the onset and offset of the flash stimulus. (**C**) Proportion of cells in each cluster (top). Mean ± SEM of the flash responses in each cluster (traces). (**D**) Schematic of the loom stimulus. (**E**) Mean ± SEM response of all of the first loom stimuli and (**F**) of the 5 consecutive loom stimuli for $Ptchd1^{Y/+}$ (black) and $Ptchd1^{Y/-}$ (orange) sSC units. (**G-I**)

Summary of mean, maximum, and time-to-peak firing (**G**, $P = 0.461$; **H**, $P = 0.523$; **I**, $P = 0.049$). (**J**) Average firing to a single loom across the 5-loom stimulus ($p = 0.758$). (**K-T**) As for (**A-J**) but for the *Cul3* mouse model. (**K**) Proportion of sSC cells that are flash responsive (*Cul3*$^{+/+}$: 0.388; *Cul3*$^{+/-}$: 0.342, $P = 0.638$, two-way $t$ test). (**L**) Sorted raster plot of neural responses to a single flash stimulus. Vertical red dotted lines indicate the onset and offset of the flash stimulus. (**M**) Proportion of cells in each cluster (top). Mean ± SEM of the flash responses in each cluster (traces). (**N**) Schematic of the loom stimulus. (**O**) Mean ± SEM response of all of the first loom stimuli and (**P**) of the 5 consecutive loom stimuli for *Cul3*$^{+/+}$ (black) and *Cul3*$^{+/-}$ (magenta) sSC units. (**Q-S**) Summary of mean, maximum, and time-to-peak firing (**Q**, $P = 0.222$; **R**, $P = 0.214$; **S**, $P = 0.879$). (**T**) Average firing to a single loom across the 5-loom stimulus ($p = 0.267$). *Ptchd1*$^{Y/+}$, $n = 3$, 263 sSC cells; *Ptchd1*$^{Y/-}$, $n = 3$, 202 sSC cells. *Cul3*$^{+/+}$, $n = 3$, 205 sSC cells; *Cul3*$^{+/-}$, $n = 3$, 204 sSC cells. *P*-values: Wilcoxon's test, *p*-values: two-way repeated measures ANOVA. The data underlying this figure can be found in S14 Data.
(TIF)

**S6 Fig. Optogenetic controls. (A, B)** Mean speed responses to light stimulation at increasing laser intensities in *Setd5*$^{+/+}$;VGluT2-Cre (**A**) and *Setd5*$^{+/-}$;VGluT2-Cre (**B**) mice injected with AAV-GCaMP6m. Blue shaded areas show 1 s of 10 Hz light stimulation. (**C**) Change in speed upon light activation at different laser intensities for *Setd5*$^{+/+}$ (top, $n = 1$. 0.01 mW mm$^{-2}$: $P = 0.700$, 3 trials; 0.5 mW mm$^{-2}$: $P = 0.472$, 10 trials; 5.0 mW mm$^{-2}$: $P > 0.995$, 5 trials; 10.0 mW mm$^{-2}$: $P = 0.700$, 3 trials; 15.0 mW mm$^{-2}$: $P = 0.667$, 2 trials; 20.0 mW mm$^{-2}$: $P = 0.442$, 8 trials) and *Setd5*$^{+/-}$ (bottom, $n = 1$. 0.01 mW mm$^{-2}$: $P = 0.343$, 4 trials; 0.5 mW mm$^{-2}$: $P = 0.796$, 9 trials; 5.0 mW mm$^{-2}$: $P = 0.279$, 4 trials; 10.0 mW mm$^{-2}$: $P = 0.200$, 4 trials; 15.0 mW mm$^{-2}$: $P = 0.887$, 4 trials; 20.0 mW mm$^{-2}$: $P = 0.678$, 10 trials) trials. S$_{at}$ is the mean speed of the animal ±50 ms of laser onset, and S$_{im}$ is the mean speed of the animal 300-800ms after laser onset. (**D**) Proportion of trials where the speed of the mouse increases (white: S$_{im}$ > S$_{at}$ by more than 1 SD), decreases (black: S$_{im}$ < S$_{at}$ by more than 1 SD) or does not change (grey: S$_{im}$ less than 1 SD different from S$_{at}$). 0.01 mW mm$^{-2}$: 7 trials, $p = 0.165$; 0.5 mW mm$^{-2}$: 19 trials, $p = 0.624$; 5 mW mm$^{-2}$: 9 trials, $p = 0.852$; 10 mW mm$^{-2}$: 7 trials, $p = 0.766$; 15 mW mm$^{-2}$: 6 trials, $p = 0.349$; 20 mW mm$^{-2}$: 18 trials, $p = 0.815$, X$^2$ test of independence. (**E**) No relationship between escape probability and laser power (*Setd5*$^{+/+}$, $r = -0.005$, $p = 0.955$; *Setd5*$^{+/-}$, $r = -0.047$, $p = 0.577$). (**F**) No relationship between maximum speed and laser power (*Setd5*$^{+/+}$, $r = -0.014$, $p = 0.871$; *Setd5*$^{+/-}$, $r = 0.011$, $p = 0.8986$). (**G**) No relationship between the Log$_{10}$(Immediate ΔSpeed) and laser power (*Setd5*$^{+/+}$, $r = -0.057$, $p = 0.496$; *Setd5*$^{+/-}$, $r = 0.283$, $p = 0.149$). (**H**) Confocal micrographs of AAV-DIO-ChR2 viral expression pattern at different anterior-posterior (AP) positions and optic fibre placement above the idSC, coordinates are in mm and from bregma. (**I, J**) Close-up to the dPAG in *Setd5*$^{+/+}$ and *Setd5*$^{+/-}$, respectively (scale bar: 200 μm). *p*-values: Pearson's correlation, *P*-values: paired Wilcoxon's tests, unless specified. *Setd5*$^{+/+}$; VGluT2-Cre, $n = 1$, 147 trials; *Setd5*$^{+/-}$; VGluT2-Cre, $n = 1$, 145 trials. The data underlying this figure can be found in S15 Data.
(TIF)

**S7 Fig. Supporting ex vivo slice electrophysiological analysis. (A)** Location of dPAG and dmSC cells that were recorded and filled with biocytin (scale bar: 100 μm). Insets show close-ups of the biocytin-filled dPAG cells (top, scale bar: 50 μm) and dmSC cells (bottom, scale bar: 20 μm). (**B-E**) Spontaneous neurotransmission in dPAG (*Setd5*$^{+/+}$: $n = 6$, 19 cells; *Setd5*$^{+/-}$: $n = 7$, 21 cells). (**B**) Average sEPSC frequency (*Setd5*$^{+/+}$: 7.8 Hz; *Setd5*$^{+/-}$: 5.0 Hz, $P = 0.7743$); (**C**) average sEPSC amplitude (*Setd5*$^{+/+}$: 16.0 pA; *Setd5*$^{+/-}$: 14.0 pA, $P = 0.704$); (**D**) average sEPSC rise time (*Setd5*$^{+/+}$: 1.50 ms; *Setd5*$^{+/-}$: 1.60 ms, $P = 0.948$); (**E**) average sEPSC decay time

($Setd5^{+/+}$: 9.64 ms; $Setd5^{+/-}$: 10.5 ms, $P = 0.838$). (**F**) Firing frequency at baseline splits the recorded cells into 2 groups. (Average baseline frequency: $Setd5^{+/+}$ high frequency, 5 cells, 123.5 Hz; $Setd5^{+/-}$ high frequency, 6 cells, 122.2 Hz; $Setd5^{+/+}$ low frequency, 13 cells, 1.12 Hz; $Setd5^{+/-}$ low frequency, 14 cells, 1.00 Hz). (**G**) Proportion of recorded cells from $Setd5^{+/+}$ (left) and $Setd5^{+/-}$ (right) mice that exhibit high frequency (active) or inactive at baseline. (**H**) Top, example traces from an active (high frequency firing), putative GABAergic, cell (left) and an inactive, putative glutamatergic, cell (right) at baseline. Centre top, mean current-firing relationship of all putative GABAergic (left) and putative glutamatergic (right) $Setd5^{+/-}$ cells in red. Light pink lines represent individual cells. Centre bottom, mean current-firing relationship of all putative GABAergic (left) and putative glutamatergic (right) $Setd5^{+/+}$ cells in black. Light grey lines represent individual cells. Bottom, summary of the relationship between current injection and action potential firing for putative GABAergic (left, $P = 0.560$ for the effect of genotype) and glutamatergic (right, $P < 0.001$ for the effect of genotype) cells. The grey area indicates the current injection values that are significantly different between $Setd5^{+/+}$ cells (black) and $Setd5^{+/-}$ cells (red) found by a multiple comparisons analysis with Tukey correction. (**I**) Average shape (top) and phase plane analysis (bottom) of the action potentials generated in the rheobase sweep of putative GABAergic ($Setd5^{+/+}$, 5 cells, 66 spikes; $Setd5^{+/-}$, 6 cells, 55 spikes) and putative glutamatergic ($Setd5^{+/+}$, 13 cells, 42 spikes; $Setd5^{+/-}$, 14 cells, 72 spikes) cells. (**J**) Action potential kinetics analysis of putative GABAergic cells. Left to right: action potential amplitude ($V_{max}$) ($Setd5^{+/+}$, 66 spikes, 27.1 mV; $Setd5^{+/-}$, 55 spikes, 33.3 mV, $p = 0.091$), width at half-peak ($Setd5^{+/+}$, 66 spikes, 0.877 ms; $Setd5^{+/-}$, 55 spikes, 0.890 ms, $p = 0.447$); action potential threshold ($Setd5^{+/+}$, 66 spikes, −35.7 mV; $Setd5^{+/-}$, 55 spikes, −37.8 mV, $p = 0.009$). (**K**) Summary of the relationship between current injection and action potential firing for the dPAG cells that receive proven dmSC input as defined by the optogenetic experiments of Fig 5A and 5B, showing a strong reduction in firing ($p < 0.001$) $Setd5^{+/+}$, $n = 3$, 9 cells; $Setd5^{+/-}$, $n = 4$, 12 cells. The grey area indicates the current injection values that are significantly different between the genotypes. (**L**) Left, rheobase current ($Setd5^{+/+}$: 19 dPAG cells, 13.7 pA; $Setd5^{+/-}$: 21 dPAG cells, 22.1 pA, $P = 0.609$). Right, soma size ($Setd5^{+/+}$: 19 dPAG cells, 135.3 μm; $Setd5^{+/-}$: 21 dPAG cells, 139.6 μm, $P = 0.899$, two-tailed $t$ test.) (**M**) Left, rheobase current ($Setd5^{+/+}$: 15 dmSC cells, 14.0 pA; $Setd5^{+/-}$: 15 dmSC cells, 14.0 pA, $P > 0.999$, two-tailed $t$ test). Right, soma size ($Setd5^{+/+}$: 9 dmSC cells, 139.9 μm; $Setd5^{+/-}$: 9 dmSC cells, 129.6 μm, $P = 0.609$, two-tailed $t$ test). (**N-R**) Intrinsic properties of dmSC cells ($Setd5^{+/+}$: $n = 4$, 15 cells; $Setd5^{+/-}$: $n = 4$, 15 cells). (**N**) Summary of the relationship between current injection and action potential firing in dmSC cells ($Setd5^{+/+}$, black, $n = 15$; $Setd5^{+/-}$, red, $n = 15$, $P = 0.066$ for a main effect of genotype, two-way repeated measures ANOVA). (**O**) Input resistance ($Setd5^{+/+}$: 1.29 GΩ; $Setd5^{+/-}$: 1.56 GΩ, $P = 1721$). (**P**) Tau ($Setd5^{+/+}$: 54.5 ms; $Setd5^{+/-}$: 66.4 ms, $P = 0.229$). (**Q**) Membrane capacitance ($Setd5^{+/+}$: 44.4 pF; $Setd5^{+/-}$: 44.7 pF, $P = 0.961$). (**R**) Resting membrane potential ($Setd5^{+/+}$: −62.3 mV; $Setd5^{+/-}$: −62.0 mV, $P = 0.795$). $p$-values: Wilcoxon's tests, $P$-values: repeated-measures analysis of variance (ANOVA), unless specified. In **H, I, K,** and **N**, shaded areas represent SEM. The data underlying this figure can be found in S16 Data.
(TIF)

**S8 Fig. Kv protein level analysis in the superior colliculus and cortex. (A, B)** Volcano plots for differential protein levels in the superior colliculus (SC) and cortex (CTX) between adult $Setd5^{+/+}$ and $Setd5^{+/-}$ mice ($n = 6$ independent samples per genotype and brain area, cyan dots represent annotated proteins as ion-channels, and purple dots represent Kv1.1, Kv1.2, and Kv1.6, horizontal dashed line represents the significance threshold ($p$-value $< 0.1$, two-sided moderated $t$ test), vertical dashed lines indicate fold change value of 0.4 between $Setd5^{+/+}$ and

$Setd5^{+/}$. (**C**) Tissue-specific western blots of Kv1.1 protein content in the SC and the CTX for $Setd5^{+/+}$ ($n = 3$) and $Setd5^{+/-}$ mice ($n = 3$) and (**D**) their quantification (SC: $Setd5^{+/+}$, 0.210; $Setd5^{+/-}$, 0.275, $P = 0.226$. CTX: $Setd5^{+/+}$, 0.167; $Setd5^{+/-}$, 0.235, $P = 0.392$). *P*-values are two-tailed Wilcoxon's signed-rank test. The data underlying this figure can be found in S17 Data and S7 Data.
(TIF)

**S9 Fig. Electrophysiological *Ptchd1* and *Cul3* characterisation.** Intrinsic properties of dPAG cells in *Cul3* and *Ptchd1* animals. (**A, J**) Input resistance (**A**, $Ptchd1^{Y/+}$, 1.03 GΩ; $Ptchd1^{Y/-}$, 1.20 GΩ, $p = 0.623$; **J**, $Cul3^{+/+}$, 1.12 GΩ; $Cul3^{+/-}$, 1.14 GΩ, $p = 0.896$). (**B, K**) Membrane capacitance (**B**, $Ptchd1^{Y/+}$, 55.6 pF; $Ptchd1^{Y/-}$, 48.2 pF, $p = 0.299$; **K**, $Cul3^{+/+}$, 66.6 pF; $Cul3^{+/-}$, 70.8 pF, $p = 0.896$). (**C, L**) Membrane constant tau (**C**, $Ptchd1^{Y/+}$, 52.0 ms; $Ptchd1^{Y/-}$, 51.2 ms, $p = 0.795$; **L**, $Cul3^{+/+}$, 63.9 ms; $Cul3^{+/-}$, 74.3 ms, $p = 0.212$). (**D, M**), Resting membrane potential (**D**, $Ptchd1^{Y/+}$, −63.5 mV; $Ptchd1^{Y/-}$, −63.5 mV, $p = 0.749$; **M**, $Cul3^{+/+}$, −64.0 mV; $Cul3^{+/-}$, −63.5 mV, $p = 0.844$). Action potential kinetics for all spikes generated in the rheobase sweep. (**E, N**) Action potential amplitude ($V_{max}$) (**E**, $Ptchd1^{Y/+}$, 33.5 mV; $Ptchd1^{Y/-}$, 33.6 mV, $p = 0.092$; **N**, $Cul3^{+/+}$, 43.0 mV, $Cul3^{+/-}$, 41.9 mV, $p = 0.984$). (**F, O**), After-hyperpolarisation (AHP) amplitude (**F**, $Ptchd1^{Y/+}$, −63.3 mV; $Ptchd1^{Y/-}$, −60.7 mV, $p < 0.001$; **O**, $Cul3^{+/+}$, −63.8 mV, $Cul3^{+/-}$, −64.4 mV, $p = 0.296$). (**G, P**) Width at half-peak (**G**, $Ptchd1^{Y/+}$, 0.580 ms; $Ptchd1^{Y/-}$, 0.817 ms, $p < 0.001$; **P**, $Cul3^{+/+}$, 0.717 ms, $Cul3^{+/-}$, 0.736 ms, $p = 0.565$). (**H, Q**) Rise time (**H**, $Ptchd1^{Y/+}$, 0.635 ms; $Ptchd1^{Y/-}$, 0.734 ms, $p = 0.011$; **Q**, $Cul3^{+/+}$, 0.790 ms, $Cul3^{+/-}$, 0.784 ms, $p = 0.399$). (**I, R**) Exponential decay constant (**I**, $Ptchd1^{Y/+}$, 0.066; $Ptchd1^{Y/-}$, 0.043, $p < 0.001$; **R**, $Cul3^{+/+}$, 0.052, $Cul3^{+/-}$, 0.067, $p = 0.867$). Points represent $Ptchd1^{Y/+}$ or $Cul3^{+/+}$ (grey) and $Ptchd1^{Y/-}$ (orange) or $Cul3^{+/-}$ (pink) dPAG cells. $Ptchd1^{Y/+}$, $n = 4$, (**A-D**) 16 cells, (**E-I**) 58 spikes; $Ptchd1^{Y/-}$, $n = 4$, (**A-D**) 16 cells, (**E-I**) 72 spikes. $Cul3^{+/+}$, $n = 3$, (**J-M**) 12 cells, (**N-R**) 68 spikes; $Cul3^{+/-}$, $n = 3$, (**J-M**) 12 cells, (**N-R**) 73 spikes. (**S**) Summary of the relationship between current injection and action potential firing showing a strong reduction in firing in $Ptchd1^{y/-}$ dPAG cells. Effect of genotype without **α**-DTX: $P = 0.046$; effect of **α**-DTX on $Ptchd1^{Y/-}$ firing: $P = 0.680$. ($Ptchd1^{Y/+}$, $n = 4$, 16 cells; $Ptchd1^{Y/-}$, $n = 4$, 20 cells; $Ptchd1^{Y/-}$ with **α**-DTX, $n = 3$, 13 cells). Inset, representative example traces to a 50-pA current injection for $Ptchd1^{Y/+}$ (black, top) and $Ptchd1^{Y/-}$ (orange, bottom). (**T**) Average shape (left) and phase plane analysis (right) of the action potentials generated in the rheobase sweep ($Ptchd1^{Y/+}$, 16 cells, 58 spikes; $Ptchd1^{Y/-}$, 20 cells, 72 spikes). (**U**) Summary of the relationship between current injection and action potential firing in *Cul3* dPAG cells ($Cul3^{+/+}$, grey, 12 cells, $n = 3$; $Cul3^{+/-}$, pink, 12 cells, $n = 3$, $P = 0.683$ for the effect of genotype). (**V**) Average shape (left) and phase plane analysis (right) of the action potentials generated in the rheobase sweep for $Cul3^{+/+}$ (grey, 68 spikes) and $Cul3^{+/-}$ (pink, 73 spikes). Box-and-whisker plots show median, IQR, and range. Shaded areas represent SEM. Lines are shaded areas, mean ± SEM, respectively. *p*-values are Wilcoxon's rank-sum tests, and *P*-values are two-way repeated measures ANOVA, unless specified. The data underlying this figure can be found in S18 Data.
(TIF)

**S1 Video. Example looming escape responses across ASD models.**
(MOV)

**S2 Video. Example repetitive looming escape response in Setd5 mice.**
(MOV)

**S3 Video. Example α-DTX rescue experiments.**
(MOV)

**S1 Data. Numerical data for Fig 1C–1K.**
(XLSX)

**S2 Data. Numerical data for Fig 2A–2F and 2H.**
(XLSX)

**S3 Data. Numerical data for Fig 2A–2P.**
(XLSX)

**S4 Data. Numerical data for Fig 2D–2G, 4H, 4I, 4K and 4L.**
(XLSX)

**S5 Data. Numerical data for Fig 2C–2F and 2G–2I.**
(XLSX)

**S6 Data. Numerical data for Fig 6C.**
(XLSX)

**S7 Data. Raw western blot images from Figs 1 and S8.**
(PDF)

**S8 Data. Numerical data for Fig 7C–7H.**
(XLSX)

**S9 Data. Numerical data for S1A–S1N Data.**
(XLSX)

**S10 Data. Numerical data for S2A–S2F Fig.**
(XLSX)

**S11 Data. Numerical data for S3D–S3Q Fig.**
(XLSX)

**S12 Data. Raw qPCR images from S3 Fig.**
(PDF)

**S13 Data. Numerical data for S4D, S4F–S4I, S4K–S4P Fig.**
(XLSX)

**S14 Data. Numerical data for S5A–S5C, S5F–S5M Fig.**
(XLSX)

**S15 Data. Numerical data for S6A–S6G Fig.**
(XLSX)

**S16 Data. Numerical data for S7B–S7F and S7H–S7M Fig.**
(XLSX)

**S17 Data. Numerical data for S8A Fig.**
(XLSX)

**S18 Data. Numerical data for S9A–S9V Fig.**
(XLSX)

## Acknowledgments

We thank Armel Nicolas, Bella Bruszel, and Ewelina Dutkiewicz from the ISTA Mass Spectrometry Service (Lab Services Facilities) for all Proteomics work, including samples

preparation, LC/MS data acquisition, searches, and data evaluation. We thank Prof. Peter Jonas for his suggestion on the involvement of potassium channels and members of the Neuroethology group for their comments on the manuscript; Katalin Szigeti and Julie Murmann for experimental help. This research was supported by the Scientific Service Units of IST Austria through resources provided by the Lab Support Facility, the Imaging and Optics Facility, the Machine Shop Unit, and the Preclinical Facility, especially Freyja Langer and Michael Schunn.

## Author Contributions

**Conceptualization:** Laura E. Burnett, Ryuichi Shigemoto, Maximilian Joesch.

**Data curation:** Laura E. Burnett.

**Formal analysis:** Laura E. Burnett, Peter Koppensteiner, Olga Symonova, Tomás Masson.

**Funding acquisition:** Maximilian Joesch.

**Investigation:** Laura E. Burnett, Peter Koppensteiner, Tomás Masson.

**Methodology:** Laura E. Burnett, Peter Koppensteiner, Olga Symonova, Tomás Masson, Tomas Vega-Zuniga, Ximena Contreras, Thomas Rülicke, Ryuichi Shigemoto, Gaia Novarino, Maximilian Joesch.

**Project administration:** Laura E. Burnett, Maximilian Joesch.

**Resources:** Ximena Contreras, Thomas Rülicke, Ryuichi Shigemoto, Gaia Novarino, Maximilian Joesch.

**Software:** Olga Symonova.

**Supervision:** Tomas Vega-Zuniga, Gaia Novarino, Maximilian Joesch.

**Visualization:** Laura E. Burnett, Tomás Masson, Maximilian Joesch.

**Writing – original draft:** Laura E. Burnett, Maximilian Joesch.

**Writing – review & editing:** Laura E. Burnett, Maximilian Joesch.

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
