## [Editor Report · Decision Letter 0]

16 Jan 2024

Dear Dr Joesch, 

Thank you for submitting your manuscript entitled "Recurrent impairments in visual perception and place avoidance across autism models are causally linked in the haploinsufficiency model of intellectual disability Setd5" for consideration as a Research Article by PLOS Biology.

Your manuscript has now been evaluated by the PLOS Biology editorial staff as well as by an academic editor with relevant expertise and I am writing to let you know that we would like to send your submission out for external peer review.

Once your full submission is complete, your paper will undergo a series of checks in preparation for peer review. After your manuscript has passed the checks it will be sent out for review. To provide the metadata for your submission, please Login to Editorial Manager (https://www.editorialmanager.com/pbiology) within two working days, i.e. by Jan 18 2024 11:59PM.

Kind regards,

Lucas

Lucas Smith, Ph.D.

Senior Editor

PLOS Biology

lsmith@plos.org

---

## [Decision Letter · Decision Letter 1]

13 Feb 2024

Dear Dr Joesch,

Thank you for your patience while your manuscript "Recurrent impairments in visual perception and place avoidance across autism models are causally linked in the haploinsufficiency model of intellectual disability Setd5" was peer-reviewed at PLOS Biology. It has now been evaluated by the PLOS Biology editors, an Academic Editor with relevant expertise, and by several independent reviewers. 

In light of the reviews, which you will find at the end of this email, we would like to invite you to revise the work to thoroughly address the reviewers' reports.

As you will see below, while the reviewers appreciate that the topic of the study is interesting and that the current experiments are well done, the each highlight that additional work is needed to fully support and flesh out the conclusions of the study to the level needed for publicaiton at PLOS Biology. Reviewer 1 has highlighted that additional circuit specific modulation and bidirectional stimulation are needed to strengthen circuit level understanding, while Reviewer 2 suggests that more work is needed to strengthen the molecular and synaptic aspects of the study - and we think these points should be experimentally addressed. 

Given the extent of revision needed, we cannot make a decision about publication until we have seen the revised manuscript and your response to the reviewers' comments. Your revised manuscript is likely to be sent for further evaluation by all or a subset of the reviewers.

We expect to receive your revised manuscript within 3 months- however we are happy to grant an extension if needed. Please email us (plosbiology@plos.org) if you have any questions or concerns, or would like to request an extension. 

**IMPORTANT - SUBMITTING YOUR REVISION**

*Re-submission Checklist*

*Published Peer Review*

*PLOS Data Policy*

*Blot and Gel Data Policy*

Sincerely,

Luke

Lucas Smith, Ph.D.

Senior Editor

PLOS Biology

lsmith@plos.org

REVIEWS:

Reviewer #1: In this study by Laura E. Burnett et al., the authors explore a recurrent impairment in visual threat perception that is similarly affected in three independent models of Autism Spectrum Disorder (ASD) with different molecular etiologies. They observe a Kv-channel induced dPAG hyperexcitability phenotype, suggesting that the integration and action initiation necessary for adequate perceptual decision-making are disrupted in Setd5 mice. While the topic is intriguing and potentially significant, there are several concerns that need addressing.

Major:

1. The authors assume that the dPAG is a key downstream brain region for the dmSC to generate robust escape responses. Therefore, they should label the postsynaptic neurons in the dPAG that receive innervation from the SC using viral tracing tools and then conduct whole-cell patch-clamp recordings. Moreover, implementing pathway-specific modulation using the appropriate viruses would be advantageous.

2. Figure 4: the authors employ optogenetic stimulation to activate SC neurons, successfully reproducing the delayed LER phenotype in Setd5+/- animals. To comprehensively elucidate the role of the SC, bilateral optogenetic activation is crucial. Additionally, providing a detailed, high-definition confocal micrograph of the injection sites, along with a clear depiction of the virus expression pattern, would be beneficial.

3. Figure 5: the referenced transcriptomic data suggests upregulation of Setd5+/- Kv channels, and the application of -DTX reversed the hypoexcitability phenotype in Setd5+/- cells. However, the authors found no changes in Kv channels within the PAG. The authors attribute this discrepancy to homeostatic misregulation but fail to provide supporting references.

4. Figure 6: the authors attempt to mitigate delayed LER and place avoidance by implanting a drug delivery cannula in close proximity to the dPAG. However, considering the superior location of the SC to the PAG, how can one ensure the prevention of any potential damage to the SC?

5. The uppercase and lowercase letters in the result figures within the manuscript do not correspond to those in the figure captions, causing considerable confusion.

Reviewer #2: The manuscript by Burnett et al. has provided well-controlled behavioral data on the conserved impairment of visual perception and place avoidance in 3 autism models (Setd5+/-, Cul3+/- and Ptchd1Y/-). In addition, it has revealed that the behavioral deficits are causally linked to a potassium channel (Kv1)-mediated hypoexcitability in dorsal periaqueductal grey (dPAG) of Setd5+/- mice. While the experiments are well designed and executed, there are a few concerns that need to be addressed.

1. The identified molecular mechanism (Kv1-mediated hypoexcitability of dPAG neurons) does not apply to Cul3+/- and Ptchd1Y/- mice, suggesting that it is not a common basis of perceptual impairment across autism models. The title is confusing and misleading, and should be changed. The abstract should clarify that the mechanism is limited to only one autism model. 

2. More discussions on the different mechanisms for the conserved LER deficits across autism models are needed. 

3. Fig. 5K does not clearly indicate the significant alteration of KCNA1 and KCNA2 in Setd5+/- mice. Are they "differential protein levels" (Line 117) or differential mRNA levels? Are they from the same kind of Setd5+/- mice as what is used here? What cell types were used in Fig. 5K? The authors need to confirm the changes in the expression of Kv1.1, Kv1.2, Kv1.6 and other K channels (targets of a-DTX) in pPAG cells of Setd5+/- mice. 

4. The conclusion will be more convincing if the authors perform patch clamp recordings to compare Kv1.1 channel currents in pPAG cells of Setd5+/+ vs. Setd5+/- mice.

5. Fig. 5G shows no effect of a-DTX on firing rate of Setd5+/+ mice, why is that? Blocking K+ channels is expected to increase firing rate in WT mice too. 

6. Line 213, Fig. 4j should be changed to Fig. S4j. Line 215, Fig. 4o,p should be changed to Fig. S4o,p. Line 251, Fig. 4 should be changed to Fig. 5.

---

## [Decision Letter · Decision Letter 2]

15 Apr 2024

Dear Dr Joesch,

Thank you for your patience while we considered your revised manuscript "Common behavioural impairments in visual perception and place avoidance across autism models are causally linked in the haploinsufficiency model of intellectual disability Setd5" for publication as a Research Article at PLOS Biology. Your revised study has been evaluated by the PLOS Biology editors, the Academic Editor and the original reviewers. 

The reviews are appended below. As you will see, while reviewer 1 is satisfied by the revision, reviewer 2 has a number of lingering concerns. After discussion with the Academic Editor, we think these last issues identified by reviewer 2 are important and that they would need to be thoroughly addressed before we can consider your study for publication. We note that this will likely require the generation of additional data to bolster your conclusions. While we tend to be hesitant to invite a second round of experimental revision, given our interest in the study and the limited scope of the remaining work required, we are willing to invite one last revision to address the reviewer comments. 

Given the extent of revision needed, we cannot make a decision about publication until we have seen the revised manuscript and your response to the reviewers' comments. Your revised manuscript may be sent for further evaluation by all or a subset of the reviewers.

**IMPORTANT - SUBMITTING YOUR REVISION**

*Re-submission Checklist*

*Published Peer Review*

*PLOS Data Policy*

*Blot and Gel Data Policy*

Sincerely,

Luke

Lucas Smith, Ph.D.

Senior Editor

PLOS Biology

lsmith@plos.org

REVIEWS:

Reviewer #1, Chaoran Ren (note, reviewer 1 has signed this review): The authors have addressed my concerns. This reviewer has no further comments.

Reviewer #2: The revised manuscript has addressed some concerns. One improvement is to add patch clamp recordings to compare Kv currents in pPAG cells of Setd5+/+ vs. Setd5+/- mice (new Fig. 5I). However, a few key points were not adequately addressed, which raises doubts on the validity of some of the main conclusions. 

1. In my last review, I pointed out that the identified molecular mechanism (Kv1-mediated hypoexcitability of dPAG neurons) does not apply to Cul3+/- and Ptchd1Y/- mice, suggesting that it is not a common basis of perceptual impairment across autism models, so the original title (Recurrent impairments in visual perception and place avoidance across autism models are causally linked in the haploinsufficiency model of intellectual disability Setd5) is misleading. Now the new title "Common behavioural impairments in visual perception and place avoidance across autism models are causally linked in the haploinsufficiency model of intellectual disability Setd5" has little change and is still problematic. While the behavioral phenotypes are similar among the three autism models, molecular mechanisms are distinct. Thus, it lacks evidence showing that they are "causally linked". The title has to be toned down.

2. In my last review, I pointed out that Fig. 5K (now Fig. 6A) does not clearly indicate the significant alteration of KCNA1 and KCNA2 in Setd5+/- mice and the authors need to confirm the changes in the expression of Kv1.1, Kv1.2, Kv1.6 (targets of a-DTX) in pPAG cells of Setd5+/- mice. Now Fig. 6B &C shows only one protein, Kv1.1 (KCNA1), which actually does not show a significant change in the volcano plot (Fig. 6A). With 3 samples in each group of the WB experiments (no labeling of group identities), the statistical conclusion (ns) is unconvincing. In addition, Fig. 6D has no quantification. Why Kv1.6 (KCNA6) was not tested? The direct evidence supporting the alteration of Kv1 channels in Setd5+/- phenotypes is lacking.

3. In my last review, I asked why Fig. 5G shows no effect of α-DTX on firing rate in WT mice, since blocking K+ channels is expected to increase firing rate. The authors responded (p16-17) that "most of the Kv channels are either not located in the initial segment of the axon" or "they have a lower conductance, e.g., if modulated (what does this mean?)". These speculations are contradictory to the finding that "Kv1.1 is involved in the homeostatic control of firing [33,34] through modulation of the axon initial segment (AIS)[35]". The result raises concerns on the conclusion that DTX-sensitive Kv1 channels cause hypo-excitability in Setd5+/- mice.

---

## [Editor Report · Decision Letter 3]

7 May 2024

Dear Max,

Thank you for the submission of your revised Research Article "Shared behavioural impairments in visual perception and place avoidance across different autism models are driven by periaqueductal grey hypoexcitability in Setd5 haploinsufficient mice" for publication in PLOS Biology. We are satisfied by your responses to the last reviewer concerns and with the underlying data provided in response to our editorial requests. Therefore, on behalf of my colleagues and the Academic Editor, Eunjoon Kim, I am pleased to say that we can in principle accept your manuscript for publication, provided you address any remaining formatting and reporting issues. These will be detailed in an email you should receive within 2-3 business days from our colleagues in the journal operations team; no action is required from you until then. Please note that we will not be able to formally accept your manuscript and schedule it for publication until you have completed any requested changes.

**IMPORTANT: A few notes 

1 - thank you for providing an updated S1_raw images file, containing the fully annotated uncropped western blot images supporting your study, over email. I have swapped the most recent version that you provided with the previous version (which was missing some annotations). 

 >>Please do double check your manuscript file inventory and make sure that this change looks good on your end. 

2 - Similarly, thank you for providing the underlying data for your study as a deposition to PRIDE and as supplemental excel files. I have updated your 'data availability statement' in our online system to reference this data, and have uploaded the excel files to our system. 

 >>Again, please double check that these changes look good on your end. 

 >>Please note, that when changing your data availability statement, I included the reviewer access code for the proteomics data that you provided me over email. However, this dataset will need to be made publicly available before publication and so the DAS should be changed accordingly when that change is made. 

 >>Please add, to each figure legend (including supplemental), a sentence that references the relevant underlying data file. For example, you can add the sentence "the data underlying this figure can be found in ___". 

3 - As discussed over email, please update your abstract to indicate that the study was done in mice. (I made this change in our online system, but please also update your manuscript file). 

PRESS

Sincerely, 

Lucas Smith, Ph.D.

Senior Editor

PLOS Biology

lsmith@plos.org